

# Long-distance electron transport occurs globally in marine sediments

Laurine D.W. Burdorf[1], Anton Tramper[1], Dorina Seitaj[1,2], Lorenz Meire[3,4], Silvia Hidalgo-Martinez[1], Eva-Maria Zetsche[1,5], Henricus T.S. Boschker[1], Filip J.R. Meysman[1,2]

[1] NIOZ Royal Netherlands Institute for Sea Research and Utrecht University, Korringaweg 7, 4401 NT Yerseke, the Netherlands
[2] Department of Analytical, Environmental and Geo-Chemistry, Vrije Universiteit Brussel, Pleinlaan 2, 1050 Brussel, Belgium
[3] Greenland Institute of Natural Resources, Greenland Climate Research Centre, P. O. Box 570, Kivioq 5, 3900 Nuuk, Greenland
[4] Arctic Research Centre, Aarhus University, 8000 Aarhus, Denmark
[5] Department of Marine Sciences, University of Gothenburg, Carl Skottsberg gata 22B, 41319 Gothenburg, Sweden

*Correspondence to*: Laurine D.W. Burdorf (laurine.burdorf@nioz.nl)

**Abstract** Recently, long filamentous bacteria have been reported to conduct electrons over centimetre distances in marine sediments. These so-called cable bacteria perform a novel "electrogenic" form of sulfur oxidation, whereby long-distance electron transport links sulfide oxidation in deeper sediment horizons to oxygen reduction in the upper millimetres of the sediment. Electrogenic sulfur oxidation exerts a strong impact on the sediment biogeochemistry, but it is currently unknown how prevalent the process is within the seafloor. Here we provide a state-of-the-art assessment of its global distribution by combining new field observations with previous reports from literature. This synthesis demonstrates that electrogenic sulfur oxidation mediated by long-distance electron transport is a widespread phenomenon in the present-day seafloor. The process is found in different oceanographic regions and climate zones (The Netherlands, Greenland, USA, Australia), and thrives in a range of different coastal habitats (estuaries, salt marshes, mangroves, coastal hypoxic basins, intertidal flats). The combination of a widespread occurrence and a strong local geochemical imprint suggests that electrogenic sulfur oxidation could be an important, and hitherto overlooked, component of the marine cycle of carbon, sulfur and other elements.

**Keywords** *Long-distance electron transport, cable bacteria, electrogenic sulfur oxidation, geomicrobiology*

## 1 Introduction

Recently, a novel type of filamentous bacteria has been discovered (Pfeffer et al., 2012), which are capable of generating and conducting electrical currents over centimetre-scale distances (Nielsen et al., 2010). These so-called "cable bacteria" live within the upper centimetres of the seafloor (Malkin et al., 2014), and induce an electron transport from sulfidic horizons at centimetres depth to the oxic layer near the sediment-water interface. By generating such long-distance electron transport, they induce a measurable electrical field within the sediment (Risgaard-Petersen et al., 2014, 2015). This implies that the seafloor is no longer an electrically neutral deposit, but rather an electro-active environment, where both ions and organisms are



influenced by electrical fields. This finding has major ramifications for the biogeochemical cycling and microbial ecology of the seafloor (Nielsen and Risgaard-Petersen, 2015; Seitaj et al., 2015). However, as cable bacteria have only been recently discovered, it is presently unknown how prevalent long-distance electron transport is in the seafloor on a global scale. Here, we provide a first assessment of the global distribution of electro-active marine sediments.

Cable bacteria bypass the traditional redox cascade by linking two spatially segregated redox half-reactions by means of long-distance electron transport (Nielsen et al., 2010; Pfeffer et al., 2012). This way, they induce a previously unknown type of sulfur cycling in marine sediments, which is referred to as electrogenic sulfur oxidation (Meysman et al., 2015). By transferring electrons from cell-to-cell along the longitudinal axis of their multi-cellular filaments, cable bacteria are able to connect the oxidation of sulfide in the deeper parts of the sediment to the reduction of oxygen in the top millimetres of the

sediment. Figure 1 provides a schematic diagram of the metabolism of cable bacteria, which involves both anodic sulfide oxidation by cells located in deeper anoxic layers and cathodic oxygen reduction by cells located in the oxic zone of the sediment.

Electrogenic sulfur oxidation (e-SOx) has a large impact on the geochemistry of the sediment, because the process is responsible for a large part of the oxygen and sulfide consumption in the sediment (Nielsen et al., 2010). At the same time, e-

SOx induces large excursions in the porewater pH (Nielsen et al., 2010; Meysman et al., 2015). Alkaline conditions (pH > 8.5) are generated in the oxic layer as a result of proton consumption during cathodic oxygen reduction, while acidic conditions (pH < 6.5) are created in deeper anoxic layers due to proton production during anodic sulfide oxidation (Fig. 1). Through their impact on the porewater pH, cable bacteria induce a range of secondary reactions, and thus stimulate the geochemical cycling of a range of elements including iron, manganese and calcium (Rao et al., 2016b; Risgaard-Petersen et al., 2012). The low pH

in the deeper layers of the sediment induces dissolution of iron sulfides and calcium carbonates, which subsequently causes the diffusion of iron and calcium cations to the oxic layer, where iron oxides form and carbonates re-precipitate to form a rigid mineral crust on top of the sediment (Risgaard-Petersen et al., 2012). This stimulation of the mineral cycling by e-SOx also induces high effluxes of calcium and alkalinity from the sediment (Rao et al., 2016b).

Recent evidence indicates that the influence of e-SOx by cable bacteria may expand well beyond the sediment and

could also impact the water column biogeochemistry (Seitaj et al., 2015; Sulu-Gambari et al., 2016a). In particular, it has been suggested that the presence of cable bacteria may influence the ecosystem functioning of seasonal hypoxic environments through the regulation of the phosphorus cycle (Sulu-Gambari et al., 2016a), manganese cycle (Sulu-Gambari et al., 2016b) and iron cycle (Seitaj et al., 2015). More specifically, it has been proposed that cable bacteria induce the formation of a large pool of sedimentary iron (hydr)oxides in spring, which then acts as a "firewall" against the development of the sulfidic water

conditions in summer (Seitaj et al., 2015). This way, cable bacteria appear capable of influencing the elemental cycling and ecosystem functioning of coastal systems at basin scales.

The large impact of cable bacteria on the geochemistry and microbial ecology of marine sediments raises the question how prevalent and dominant these bacteria are on a global scale. Until now, only a few reports are available from a restricted number of coastal sites and habitats. The geochemical fingerprint of e-SOx (Nielsen et al., 2010), and also the cable bacteria



themselves (Pfeffer et al., 2012), were originally discovered in a laboratory enrichment experiment using sediment from the Baltic Sea (Aarhus Bay, Denmark). Subsequently, the geochemical fingerprint, together with high densities of cable bacteria, were recorded under natural conditions at three different sites in the Southern North Sea, including a salt marsh, a seasonally hypoxic basin and a coastal mud accumulation site (Malkin et al., 2014). Since then, laboratory enrichments have revealed the growth of cable bacteria in sediments from salt marshes on the Atlantic coast of North America (Larsen et al., 2015; Rao et al., 2016b), while a multi-year study has shown the seasonal reappearance of intense cable bacteria activity in Lake Grevelingen, a seasonally hypoxic basin in The Netherlands (Seitaj et al., 2015; Sulu-Gambari et al., 2016a). Just recently, the presence and activity of cable bacteria was demonstrated in temperate mangroves in Australia (Burdorf et al., 2016).

There are, however, indications that the activity of cable bacteria is likely more widespread in the seafloor than currently documented. A re-evaluation of published porewater data in combination with an examination of 16S rRNA archives (Malkin et al., 2014) suggests a far broader natural distribution, indicating that long-distance transport could be prominent in other marine habitats (e.g. bivalve beds, mid-oceanic ridges, the deep sea). The goal of this study is to provide a first synthesis of the global distribution of cable bacteria in the seafloor, reviewing the data that is already published in combination with an extensive set of new observations. This analysis suggests that e-SOx is a far more important component of the sedimentary sulfur cycle in the seafloor than previously recognized.

## 2 Material and methods

### 2.1 Screening procedures

The impact of cable bacteria on marine sediments can be assessed in two ways: (1) by demonstrating that cable bacteria are present at high abundances in the sediment, or (2) by demonstrating the metabolic activity of the cable bacteria, i.e., by revealing the impact of e-SOx on the sediment geochemistry. The verification of the presence of cable bacteria, and the subsequent quantification of their abundance, is performed by microscopic and molecular techniques (Fig. 2; right column). Alternatively, the demonstration of metabolic activity of cable bacteria is achieved by the analysis of porewater geochemistry and/or a direct measurement of the electrical potential that is generated in the sediment in association with e-SOx (Fig. 2; left column). All these techniques are reviewed in detail below. Clearly, the detection of metabolic activity implies presence, though not the other way around. Low cable bacteria abundances can be detected without noticing an impact on the sediment geochemistry. Consequently, the direct demonstration of e-SOx activity provides stronger evidence for the geochemical impact of cable bacteria on the seafloor than their presence alone. However, as we will show below, high densities of cable bacteria are typically tightly correlated with high metabolic activity, and so the investigation of cable bacteria abundance remains a useful indicator.

Up to now, the presence/activity of cable bacteria in natural sediment environments has been demonstrated in two principal ways: (1) the phenomenon is directly demonstrated under field conditions (Malkin et al., 2014), or (2) it is demonstrated by means of a laboratory induction experiment (Schauer et al., 2014; Larsen et al., 2015). In a laboratory




induction, sediment is retrieved from the field, homogenized and mixed, and the activity and/or presence of cable bacteria evaluated during the incubation period (days to weeks). Such laboratory incubations can be used as a simple, quick and rather inexpensive screening method, since direct field observations are typically more challenging and logistically complex. However, prolific growth and strong activity of cable bacteria in laboratory induction experiments do not unequivocally prove

that the process also occurs at the field site from where the incubated sediment has been retrieved. Therefore, the results of a laboratory induction procedure have a potential character, while the field assessment procedure provides direct evidence for the natural impact of cable bacteria. However, as we will show below, there is a strong correlation between success in laboratory induction and natural occurrence, which suggests that laboratory induction provides useful – albeit indicative – information on the natural distribution of cable bacteria.

In this study, we have compiled previously published data on the distribution of cable bacteria in the seafloor. Additionally, we have extended this dataset by assessing the presence and/or activity of cable bacteria in a number of geographical locations during various sampling campaigns conducted between 2011 and 2016. Besides the global geographical distribution, we examine the aspect of habitat diversity within the same geographical region. To this end, we have performed a detailed investigation of different coastal habitats (e.g. salt marshes, mud flats, bivalve reefs) within a restricted region in the

Southern North Sea. In these surveys, we combined different screening procedures (Fig. 2), assessing the presence and/or activity under laboratory and/or field conditions.

### 2.2 Field sampling

The detection of metabolic activity in the field can be done either by *in situ* microsensor profiling (e.g. Risgaard-Petersen et al., 2015) or by the retrieval of sediment cores and immediate inspection by microsensor profiling either shipboard or in the

laboratory (e.g. Malkin et al., 2014). Here we only employed the latter technique. Sediment cores from shallow, intertidal sites were collected at low tide by manual insertion of plastic core liners (different types were used at different locations; inner diameter range 36 - 60 mm). Cores from deeper areas were sampled shipboard using a gravity core sampler (60 mm internal diameter; UWITEC (Austria) or Kajak sampler, KC Denmark A/S). The time between sampling and measurement was kept as short as possible, and sediment core analysis (microsensor depth profiling or core sectioning for microscopy) was always

performed within 12 hours of core sampling. During this period, the overlying water of the subtidal sediment cores was kept oxygenated by air bubbling or by submerging the sediment cores into a large water bath filled with *in situ* bottom water. Shipboard analysis within 12 hours of core sampling appears sufficiently rapid to detect the *in situ* signal. In laboratory induction experiments, cable bacteria populations typically require 5 to 14 days before the characteristic e-SOx signal is detected by microsensor profiling (Pfeffer et al., 2012; Schauer et al., 2014; Rao et al., 2016b). While we observed that core

disturbance can eliminate the characteristic e-SOx fingerprint (i.e., false negatives are possible), the shipboard detection of e-SOx activity within a period of hours provides a positive confirmation that the activity was also present *in situ* (i.e., false positives are excluded, as e-SOx activity in laboratory conditions takes >5 days to develop). Retrieved sediment cores were





always visually inspected prior to analysis and only cores with a visually undisturbed sediment-water interface were retained for analysis (potential exclusion of false negatives).

## 2.3 Laboratory induction experiments

As noted above, laboratory induction of cable bacteria growth provides an inexpensive and indicative screening method. Here, laboratory inductions were carried out in a similar fashion as done in previous studies (Pfeffer et al., 2012; Vasquez-Cardenas et al., 2015). To remove large burrowing macrofauna, the sediment was either sieved over a 0.5 – 1 mm sieve (Nielsen et al., 2010; Malkin and Meysman, 2015) or sediment cores were left undisturbed and sealed for a longer period of time to asphyxiate the benthos (Rao et al., 2014, 2016b). The exclusion of burrowing macrofauna is thought to prevent the mechanical disturbance of the cable bacteria filament network (Malkin et al., 2014) and allows to assess the effect of cable bacteria on the sediment geochemistry without the effect of bioturbation. Asphyxiated sediment cores were used as such (i.e., no homogenization). Sieved sediments were homogenized and packed into plastic core liners (inner diameter 36 mm), adjusting the height of the sediment to the rim of the core liner (for ease of microsensor profiling). At the start of the laboratory incubation, sediment cores were placed submerged in an aquarium in a temperature-controlled climate room fixed at *in situ* temperature where possible. During laboratory inductions, the overlying water of the aquarium was continuously bubbled with ambient air to retain a fully oxygenated water column.

## 2.4 Geochemical microsensor profiling

The metabolic activity of cable bacteria imposes a distinct geochemical fingerprint on the porewater, which is revealed by a combination of oxygen ($O_2$), free sulphide ($H_2S$) and pH microsensor profiling (Nielsen et al., 2010; Meysman et al., 2015). The first characteristic feature is the presence of a wide suboxic zone, defined as a sediment layer where neither $O_2$ nor $H_2S$ is detectable, i.e., concentrations are below the detection limit of 1 µM. During the development of a cable bacteria community, it has been observed that this suboxic zone gradually expands over time as the cable bacteria filament network extends away from the sediment-water interface into the deeper sediment (Schauer et al., 2014). The depth of the suboxic zone is thus a good indication of the depth to which the cable bacteria network is present. The spatial segregation of the two redox half-reactions in e-SOx leads to a distinct pH signature in the sediment porewater (Fig. 1; Meysman et al., 2015). The anodic oxidation of sulfide at depth causes proton production, whilst conversely, the cathodic reduction of oxygen and/or nitrate in the top sediment leads to strong proton consumption. This results in a distinctive pH depth profile, with a subsurface pH maximum at the oxygen penetration depth (OPD), and a pH minimum near the sulfide appearance depth (SAD). The amplitude of the excursions in the pH profile is the combined result of the sediment buffering capacity and cable bacteria activity. The pH excursion, ΔpH, defined as the maximum pH in the oxic zone minus the minimum pH in the suboxic zone, can be used as a simple indicator to compare cable bacteria activity across sites (assuming that sediment buffering capacity is similar).

In this study, micro-sensor depth profiles of $O_2$, pH and $H_2S$ were recorded using commercial micro-electrodes (Unisense A.S. Denmark, tip sizes pH: 200 µm, $H_2S$: 100 µm, $O_2$: 50 µm) operated with a motorized micromanipulator



(Unisense A.S., Denmark). Oxygen profiles were measured separately at 50 µm or 100 µm resolution, while pH and $H_2S$ were conjointly recorded with a 200 µm resolution in the oxic zone of the sediment, and increasing step size in the deeper layers of the sediments. The sensors were calibrated by following standard calibration procedures as described previously (Malkin et al., 2014) ($H_2S$: 3 to 5 point standard curve using $Na_2S$ standards; $O_2$: 2 point calibration using 100% in air bubbled seawater

and the anoxic zone of the sediment; pH: 3 NBS standards (4, 7, 10) and TRIS buffer). The pH data are reported on the total pH scale and $\Sigma H_2S$ was calculated from $H_2S$ based on the pH measured at the same depth.

## 2.5 Electrical potential profiling

Cable bacteria impose an electron transport from the anodic cells in the anoxic zone to cathodic cells in the oxic zone (Fig. 2). The resulting electrical current creates a measurable difference in electrical potential (EP) between the surface and deeper

sediment (Revil et al., 2010), which is in the range of 0.5-2 mV in marine sediments (Risgaard-Petersen et al., 2014, 2015). This EP depth profile can be recorded at high resolution by a newly developed EP microsensor (Damgaard et al., 2014), and provides a direct confirmation of metabolic activity of cable bacteria (Risgaard-Petersen et al., 2014). The current density generated by the cable bacteria can be derived from the electrical potential measurement using Ohms Law, $J = \sigma E$, where J is the flux of electrons (current density), E is the electrical field, and $\sigma$ the electrical conductivity of the sediment. The electrical

field can be derived from the EP depth profile, either using the voltage difference between the SWI and the SAD (Risgaard-Petersen et al., 2014) or as the slope of the EP profile at the oxic-anoxic transition (Damgaard et al., 2014). The conductivity $\sigma$ of the porewater is obtained from the porewater salinity relationship using the equations by Fofonoff and Millard (1983), as implemented in the R package marelac, and is corrected for tortuosity similar to the diffusion coefficient (Damgaard et al., 2014).

Here, we employed EP microsensors built according to Damgaard et al. (2014). A custom-built sensor electrode and reference electrode (Radiometer, Denmark) were connected to a voltmeter with high impedance (MB 11mV, Microscale Measurement, The Hague, The Netherlands). Signal noise was repressed using additional capacitors (total: 470 nF) connected to the voltmeter, and the analogue output of the voltmeter was converted using a 16 bit A/D converter (Unisense, Denmark). To counter signal drift, the depth profiles were recorded in two directions: from the sediment-water interface down to deeper

layers, and subsequently in the reverse direction. From these two EP profiles, the average depth profile was calculated. The salinity of the overlying water was set to match the salinity of the porewater in order to avoid the creation of diffusion potentials, which could be mistakenly interpreted as arising from long-distance electron transport.

## 2.6 Bright-field and scanning electron microscopy

Conventional bright-field microscopy can be used as a first indication of the presence of cable bacteria in the sediment. Cable

bacteria can be recognized as thin (approx. 0.5-5 µm width), long (up to >1 cm) threads, often arranged in clumps and sticking to sediment particles. Although informative, bright-field microscopy cannot provide unambiguous confirmation that the observed filaments are indeed cable bacteria. This is because various other filamentous bacteria (e.g. cyanobacteria),





filamentous fungi or plant detritus can also be present as long, thin threads in the sediment (Godinho-Orlandi and Jones, 1981). However, scanning or transmission electron microscopy does allow positive identification of cable bacteria. This is because cable bacteria display a unique topology of the outer cell surface, which shows distinctive ridges that run in parallel along the longitudinal axis of the filaments (Pfeffer et al., 2012; Malkin et al., 2014). As this ridge pattern appears unique to cable

bacteria, it unequivocally confirms that cable bacteria are present (Fig. 2). Here, scanning electron microscopy (SEM) was performed using a Phenom Pro desktop microscope (Phenom-World B.V., The Netherlands) with a beam intensity of 10 kEV. Typically, a small drop of sediment was dispersed on a suitable carrier (aluminium stub coated with a carbon adhesive pad), and subsequently air-dried. Sometimes, cable bacteria were also picked out of the sediment as clumps, using modified Pasteur pipettes as "fishing hooks". Before imaging, the sample was gold-coated for 30 seconds providing a ~5 nm thick gold layer

(Polaron E5100 sputter coater, Van Loenen Instruments, Belgium).

### 2.7 Fluorescence *In Situ* Hybridization

Fluorescence *in situ* hybridization (FISH) is a molecular staining technique which targets specific regions of the ribosomal RNA. Once a probe successfully hybridizes with its target rRNA region, a fluorescent signal is emitted that can be visualized with epifluorescence microscopy. Probes can be designed to target different phylogenetic levels, ranging from broad groups

(e.g. EUB mix, targeting ~90% of bacteria) to specific taxa. In previous studies, the probe DSB706 (Manz et al., 1992), which targets most Desulfobulbaceae as well as Thermodesulforhabdus, has been shown to successfully hybridize with cable bacteria (e.g. Pfeffer et al., 2012; Schauer et al., 2014; Seitaj et al., 2015). Here, we performed standard FISH analysis with the DSB706 probe, following the protocol described in Schauer et al. (2014). For this, 0.5 mL of sediment was preserved in 96% ethanol and stored at -20°C. Subsamples of 100 µl were transferred in 500 µl of a 1:1 mixture of PBS/ethanol, and subsequently, 10

µl of this mixture was filtered through a polycarbonate membrane filter (type GTTP, pore size 0.2 µm, Millipore, USA). Probe hybridization was performed according to previously published protocols (Pernthaler et al., 2001, 2002).

## 3 Results

### 3.1 Proof of concept: laboratory inductions compared to field measurements

Laboratory incubations of sediments have previously been used as a simple and fast screening technique to discern whether

cable bacteria potentially thrive at a given location (Larsen et al., 2015; Burdorf et al., 2016). Previous studies have shown that geochemical fingerprints (depth profiles of $O_2$, $H_2S$ and pH) obtained during laboratory induction are comparable to those obtained from direct field observations. For example, during incubations with sediment from the seasonally hypoxic Lake Grevelingen, Rao et al. (2016b) recorded porewater chemistry that was similar to the field observations by Seitaj et al. (2015). However, in these studies, the sediment was never collected at exactly the same time and place. Here, we performed a direct

comparison between laboratory inductions and field measurements. In this methodological test, we compared both the geochemical as well as the electrical fingerprints.





Intertidal sediment was collected at low tide on 9 October 2014 near an oyster reef on the barrier island of Texel (Wadden Sea, The Netherlands). Sediment for laboratory induction was collected by scooping the first 10-15 cm of surface sediment into a container. Subsequently, intact sediment cores (n = 2) were collected nearby (at approx. 1 m distance). These field cores were immediately transferred to a nearby laboratory (~30 min, NIOZ Texel) and put in an air-bubbled aquarium

filled with seawater collected from the sampling site. Within 6 hours after sampling, pH, $H_2S$, $O_2$ and EP depth profiles were recorded within the intact sediment cores (2 replicate profiles per core). The sediment for laboratory induction was sieved and homogenized upon arrival in the laboratory and kept in an anoxic jar for 2 weeks after which the laboratory induction experiment was started (2 replicate cores) with freshly prepared artificial seawater. After 19 days of incubation, pH, $H_2S$, $O_2$ and EP depth profiles were also recorded in the incubated cores (2 depth profiles for each sensor per core).

Figure 3 shows representative depth profiles recorded during the laboratory induction (right panel) and the corresponding field observations (left panel). Both cores showed a shallow oxygen penetration depth (OPD = 0.4 ± 0.06 mm in the field versus OPD = 1.2 ± 0.1 mm in the laboratory induction) and a clear acidification of the suboxic zone (ΔpH = 2.1 ± 0.4 in the field versus ΔpH = 2.4 ± 0.1 in the laboratory induction). The suboxic zone is wider in the laboratory induction (20 ± 3.2 mm) compared to the field measurement (9 ± 3.2 mm). The expected pH maximum near the OPD was only detectable

in the laboratory induction. This aligns with other recent field observations, which all report e-SOx activity and high abundances of cable bacteria but no clear pH maximum (Seitaj et al., 2015; Burdorf et al., 2016; van de Velde et al., in review). The reasons for the absence of the pH maximum are discussed in detail in Burdorf et al. (2016). The EP depth profile has a similar shape in both cases and the overall increase in EP from the sediment oxic zone to the anoxic zone is also similar (1.4 mV increase in the laboratory incubation and 1.6 mV for the field measurement). The associated current densities are higher

in the field cores (412 mA m$^{-2}$ at salinity 30 and 12°C) than in the laboratory induction (280 mA m$^{-2}$ at salinity 28 and 18°C). This points towards a more active cable bacteria population in the field cores. Overall, however, the laboratory induction and field observations provided similar geochemical and electrical fingerprints.

## 3.2 Regional habitat diversity of cable bacteria

To assess the distribution of cable bacteria on a regional scale, we visited 15 sites within the Dutch Wadden Sea and the Rhine-

Meuse-Scheldt delta area (North Sea coast of the Netherlands and Belgium; Fig. 4). These 15 sites were visited during different sampling campaigns over the years 2012-2015 (summarized in Table 1, full details in Table S1). All data reported here are field observations. It should be noted that the sites investigated are not "ecologically homogeneous", in the sense that one site may consist of more than one habitat (e.g. a site may combine a bivalve reef and its surrounding mud flat). Therefore, each sampling point is called a "location", whereas the term "site" is reserved for a geographically distinct area. Overall, we sampled

a total of 28 locations distributed over the 15 sites.



### 3.2.1 Rhine-Meuse-Scheldt delta

The sites in the Rhine-Meuse-Scheldt delta were located within a geographically restricted area (all within 100 km), but covered a diverse range of coastal habitats (Fig. 4b). We surveyed sediments in three water bodies within the Rhine-Meuse-Scheldt delta, which each have a distinct ecology and hydrodynamics regime: (1) Lake Grevelingen: a seasonally hypoxic basin with no tides, closed off from river input, and minimal exchange with the open North sea; (2) the Eastern Scheldt: a tidal marine embayment, closed off from river input, but open to the North Sea; and (3) the Western Scheldt: a macrotidal estuary open to both river input and the North sea.

(1) Lake Grevelingen. Lake Grevelingen is an enclosed marine basin in the northern arm of the former Rhine-Meuse-Scheldt estuary. Saline conditions (29-32) are permanently maintained by the small exchange of water with the North Sea through a sluice connection. Seasonal oxygen depletion is a yearly occurring phenomenon in the deeper basins of Lake Grevelingen (i.e., the former estuarine channels). This oxygen depletion typically affects the bottom waters below 15 m, which represent about 6% of the basin area (Westeijn, 2011). An elaborate sampling campaign, involving monthly water column and sediment sampling, has recently described in detail the temporal response of the sediment geochemistry to seasonal oxygen depletion, revealing the important role of the cable bacteria population dynamics (Seitaj et al., 2015; Sulu-Gambari et al., 2016a). However, these studies all focused on a single station in the Den Osse basin at 34 m water depth.

In this study, however, we examined the broader spatial distribution of cable bacteria within the sediments of Lake Grevelingen. To this end, we surveyed the depth zone affected by hypoxia (water depths ranging from 12 to 45 m) during two seasons (March and November 2015). Sampling locations were distributed over four separate basins in Lake Grevelingen along the former main estuarine channel (Table 1 and Table S1). All sampled sites were muddy sediment with a high porosity (>0.8) and with no or only few small bioturbating fauna. All sites experienced bottom water oxygen depletion over the summer. The four locations examined in March 2015 all revealed the distinctive geochemical fingerprint of e-SOx, indicating metabolic activity of cable bacteria (Fig. 5). A wide suboxic zone was observed in all stations ($35 \pm 8$ mm), together with a clear pH maximum near the OPD and deeper pH minimum ($\Delta pH = 2.1 \pm 0.7$). The presence of cable bacteria was additionally confirmed by FISH (Fig 5, top row). In November 2015, five out of the ten locations showed the e-SOx geochemical fingerprint. In these five locations, the suboxic zone was narrower ($6.8 \pm 1.4$ mm) and the acidification of the suboxic zone was less pronounced ($\Delta pH = 1.3 \pm 0.5$) compared to March. Our data, hence, demonstrate that cable bacteria are widespread in hypoxia impacted sediments in Lake Grevelingen, particularly in spring.

(2) Eastern Scheldt. Located just south of Lake Grevelingen, the Eastern Scheldt is cut off from river inputs by dams, but maintains an open connection to the North Sea through a storm surge barrier. The intertidal area of the Eastern Scheldt is composed of four main habitats: bioturbated sand flats, mud flats, salt marshes, and bivalve reefs composed of mussels and/or oysters. A survey was conducted over the course of two days in November 2015, investigating the presence and activity of cable bacteria in these four habitats. The presence of cable bacteria in cohesive sediment habitats (mud flats, salt marshes and bivalve reefs) is discussed here, the presence in bioturbated sand flats is discussed separately below.



Five intertidal sites were investigated, and at each site, one or more locations were sampled by manual coring at low tide. Sediment cores were taken from a total of eight locations distributed over these five sites, and all cores were examined by microsensor profiling within 24 hours after collection. In six locations (including 1 mud flat, 3 salt marshes, and 2 bivalve reef habitats), the distinct geochemical fingerprint of e-SOx was present with a suboxic zone of 3-25 mm and a ΔpH ranging

from 0.8 to 2.3 (Fig. 5). A subsample from all cores was examined using light microscopy and SEM, which confirmed the presence of cable bacteria in all 6 locations having the e-SOx fingerprint. A representative geochemical profile and SEM image of cable bacteria is given in Fig. 5b and Fig. 5d. Given the relatively low sampling effort, the high percentage of locations with cable bacteria demonstrates that cable bacteria are widely present within the cohesive sediment habitats of the Eastern Scheldt.

      (3) Western Scheldt. The Western Scheldt comprises a third branch of the Rhine-Meuse-Scheldt delta, and has a very

different biogeochemistry compared to Lake Grevelingen and the Eastern Scheldt. In contrast to the two northern arms, this important shipping route to the harbour of Antwerp has remained a true estuary, which connects the Scheldt River to the North Sea. Four intertidal sites were sampled in November 2015 by manually collecting cores at low-tide (8 locations in total, see full details in Table S1). The geochemical profiles showed no indication of cable bacteria activity: no pH excursions were detected (e.g. no elevated pH in the oxic zone and no sharp decrease in pH in the anoxic zone). Ensuing microscopic

examination of the sediment by light-microscopy and SEM did not reveal any presence of cable bacteria.

### 3.2.2 Wadden Sea

On the barrier island of Texel, a survey was conducted in the Mokbaai bay at the southern tip of the island (one site, five locations sampled), and within the Cocksdorp area in the northern part of the island (3 locations sampled in total). All sites are listed in Table 1 (full information on locations in Table S1). The Mokbaai area is a sheltered bay which receives an input of

sediment and organic matter from the Wadden Sea. In the bay, current velocities are decreased towards the land-side due to the presence of a bivalve reef, which spans nearly the whole inlet of the bay. Moving from offshore to inshore, five distinct habitats are sequentially present: (1) a bivalve reef of mixed mussels and oysters, (2) a mud accumulation area behind the reef with little or no burrowing infauna, (3) an intertidal mud flat with high cockle densities (*Cerastoderma edule*), (4) sandflats dominated by the lugworm *Arenicola marina*, and (5) salt marshes. In July 2014 sediment cores from each of the five habitats

were collected (3–5 cores per habitat) and analysed within 3 hours of collection in a nearby laboratory (NIOZ – Texel). Cable bacteria activity was detected in at least one core per habitat except for the sandflats. From all the cores taken at the bivalve reef (n=4), cable bacteria activity was detected in two cores (ΔpH = 1.1 – 1.3). The geochemical fingerprint of e-SOx was also present in two out of three cores collected within the mud accumulation area behind the reef (ΔpH = 0.95 – 1.2), and in all the cores (n=3) sampled above the cockle bed, which showed high pH excursions (ΔpH = 1.7 +/- 0.3). In contrast, the geochemical

fingerprint of e-SOx was absent in all cores taken from the sandy bioturbated area (n=4). The presence of cable bacteria in this sandy sediment was further investigated with light-microscopy, but no filamentous bacteria were encountered.

      In the Cocksdorp area, an extensive bivalve reef (mussels and oysters) stretches out parallel to the coastline. Three sites were sampled in July 2014: one site was in a reef section dominated by oysters (*Crassotrea gigas*), a second site was



situated near a reef section dominated by mussels (*Mytilus edulis*) and a third site was located on the intertidal mud flat between the reef and the landside. Cable bacteria were active in both reef sections. Cable bacteria activity was detected in 3 out of 4 cores taken at the "oyster site" and in two out of four cores taken at the "mussel site". No cable bacteria activity was measured (n=3) on the intertidal mud flat. The absence of cable bacteria at this site might be explained by the large cover of decaying

macroalgae on the mud flat (approx. 60%), which was present at the time of sampling and which could essentially asphyxiate the sediment (cutting off the $O_2$ supply to cable bacteria).

### 3.3        Global distribution

In addition to the habitats in the Southern North Sea area, another five coastal habitats were investigated in widely different geographical locations: Long Island Sound, USA; Yarra River, Australia; Kobbefjord, Greenland; Etang d'Urbino (seagrass

bed), France; Sabkhet Arina (salt pan), Tunisia. Given the challenging logistics of *in situ* sampling, the potential to develop an active cable bacteria population was assessed using laboratory incubations (Table 1). In the first three sites the temporal development of the cable bacteria activity was monitored over time using geochemical profiling (Fig. 6). At the two other sites, the presence and activity of cable bacteria was investigated at one time point with geochemical profiling and SEM imaging (Fig. 7).

### 3.3.1 Depositional basin, Long Island Sound, USA

Long Island Sound (New York, USA) experiences high nutrient loadings and a seasonal stratification of the water column, which leads to a yearly recurring period of oxygen depletion in the bottom waters of the eastern and central parts of the basin (Cuomo et al., 2014). Sediment was collected using a Soutar-style box corer in August 2013 at a site in a depositional area in Port Jefferson harbour (10m depth, details in Table 1). Sediment cores were subsequently sieved, homogenised and finally

incubated at room temperature (20-21°C) in a darkened incubation tank with seawater from the field site (salinity 28). After a period of overnight equilibration, microsensor profiling (pH, $H_2S$ and $O_2$) was performed daily on triplicate sediment cores to track the development of cable bacteria activity (top row, Fig. 6). Already after 3 days of incubation, we observed an acidification of the sediment (-1.2 pH units vs. overlying water (OLW)) near the SAD (4.2 mm). The full geochemical signature of e-SOx was present after 7 days of incubation. A pH maximum evolved in the oxic zone of the sediment (+0.55 pH units vs.

OLW), a wide suboxic zone was present (8 mm) and a pH minimum (-1.7 pH units vs. OLW) was located near the SAD.

### 3.3.2 Seasonally hypoxic estuary, Yarra River, Australia

The Yarra River runs through the city of Melbourne and forms an estuary that discharges into Port Philip Bay. During dry spells and a low influx of freshwater, a "salt wedge" forms in the estuary resulting in bottom water oxygen depletion. We examined a site in the Yarra River referred to as Scotch College, which is known to experience seasonal hypoxia (e.g. Roberts

et al., 2012; Robertson et al., 2015). Sediment cores were retrieved in February 2014 using a hand-corer and transferred to a nearby laboratory (Water Studies Center, Monash University, Melbourne). Sediment was sieved and homogenized to exclude





shell material and plant debris. Newly repacked cores were reoxygenated and followed up over time (Fig. 6, middle row) to measure the development of cable bacteria activity (temperature 20-25°C, salinity 18). After 4 days, the distinct pH profile of e-SOx was present in the sediment: an alkalinisation within the oxic zone (+0.4 pH units vs. OLW) accompanied by an acidification of the deeper sediment (-1.8 pH units vs. OLW). After 7 days, a suboxic zone developed (3 mm wide) and a lower pH minimum was measured at the SAD (-2.3 pH units vs. OLW). Light microscopy confirmed the presence of long, small filamentous bacteria in the cores after 3 days of incubation. Cable bacteria presence was confirmed with sediment samples taken at the end of the incubation using FISH staining.

### 3.3.3 Subpolar fjord sediment, Kobbefjord, Greenland

Kobbefjord is a fjord near the Arctic Circle in Greenland, which is composed of a series of basins and sills. The bottom water is oxygenated year-round (>200 µmol L$^{-1}$) and has a low temperature range (-0.5 – 3°C) (Sørensen et al., 2015). A site at 110 metres depth was sampled using a Kajak gravity corer (diameter 5.3 cm). Sediment cores were incubated in the laboratory at bottom water temperature (0°C). Sediment cores were asphyxiated for three weeks, and subsequently oxygenated and tracked over 98 days with pH and O$_2$ microsensor profiling (Fig. 6, bottom row). The distinctive pH profile of e-SOx was visible after 26 days, with a clear pH peak (0.7 pH unit vs. OLW) and a distinct acidification deeper down into the sediment. The cable bacteria community remained active for a long time in the sediment cores, even after 98 days since the start of the incubation.

### 3.3.4 Seagrass bed, Corsica, France

Muddy sediment was collected on 31 August 2015 from the fringe of a seagrass bed within the Urbino coastal lagoon in eastern Corsica, France (Fig. 7a, details in Table 1 and Table S1). The coastal lagoon has a high salinity (~40 at the time of sampling, typically 30-38 (Fernandez et al., 2006)). The sediment was kept in oxygenated water in a container until the return to the home laboratory in the Netherlands. There the samples were sieved and transferred to plastic core liners (diameter 3.6 cm) and incubated for 24 days in oxygenated water bath. The geochemical measurements (data not shown) from the seagrass bed sediments showed all characteristics of an electro-active sediment, with the characteristic pH excursions (ΔpH = 1.7) and the establishment of a suboxic zone (6.2 mm). Subsequently, a subsample was visualized with SEM and FISH-staining. Under the SEM, filamentous bacteria with longitudinal ridges were present (Fig. 7b) and the FISH-staining procedure revealed cable bacteria with a diameter ranging from 0.7 – 3 µm (Fig. 7c).

### 3.3.5 Coastal salt pan, Sabkhet Arina, Tunis, Tunisia

Sediment was collected on 24 April 2015 from the border of a salt pan near Tunis, Sabkhet Arina (details in Table 1 and Table S1), when the pan was inundated. The salt pans (sabkhas) in Tunisia have a highly variable salinity depending on the balance between precipitation and freshwater discharge from land versus the evaporation of the water within the salt pan. Sediment was carefully collected in a screw-top vial (Fig. 7d) and incubated in oxygenated seawater at the home laboratory in the Netherlands. After 34 days the pH depth profile was measured in the core, which showed a decrease in the suboxic zone (ΔpH





=1.2). However, due to the lack of H$_2$S measurement or the distinctive pH peak in the top millimetres, activity of cable bacteria in this core was not sufficiently proven (data not shown). Nonetheless, the presence of cable bacteria was found by microscopy. First, a subsample of the sediment under the SEM showed long filamentous bacteria with distinctive ridges on the outer cells with a diameter between 1-2 µm. Secondly, long filamentous bacteria with 1-2 µm diameters were observed during FISH

analysis.

## 4 Discussion

### 4.1 Cable bacteria: global distribution and habitats

Ever since the geochemical signature of cable bacteria was first discovered in 2010 in a laboratory induction experiment (Nielsen et al., 2010), more and more evidence has accumulated that cable bacteria could be crucial players in the natural

elemental cycling of marine sediments (Malkin et al., 2014; Nielsen and Risgaard-Petersen, 2015). When cable bacteria are present and active in a given sediment environment, they exert a large impact on the geochemical cycling due to their strong impact on the pH depth profile (Risgaard-Petersen et al., 2012; Rao et al., 2016b). However, the life history and basic ecology of cable bacteria is only starting to be unravelled, and so only little is known about the geographical distribution as well as the habitat preferences of cable bacteria. Our synthesis here reveals that e-SOx by cable bacteria could be a globally occurring

process that is active in a wide range of marine habitats. Figure 8 provides a global overview, summarizing the new observations presented in this study together with previous reported data from literature (including gene archive data with a >97% similarity to cable bacteria and published geochemical profiles with an e-SOx signature that were not interpreted as such at the time of publication). In the following paragraphs, we provide a short discussion of the different locations and habitats that harbour cable bacteria.

Cable bacteria are active in a wide range of coastal sediments, both subtidal as well as intertidal. The coastal habitats where e-SOx activity has now been documented include mud flats, fjord sediments, bivalve reefs, salt marshes, seagrass beds, mangroves, salt flats and seasonally hypoxic sediments (Fig. 8). These environments are all characterized by the accumulation of fine-grained sediments with a relatively high organic carbon content, and hence, they appear to fit the "ideal e-SOx site" as originally proposed in Malkin et al. (2014): [1] the electron donor (sulfide) is available in large quantities/concentrations

through high rates of sulfate reduction, [2] an oxygenated overlying water column supplies a favourable electron acceptor (oxygen) by diffusion into the surface layer of the sediment, and [3] sediment fauna is largely absent so that bioturbation pressure is low and the filament network is not disrupted by mechanical disturbance (new observations do find cable bacteria in bioturbated sites, see discussion in paragraph 4.3).

### 4.1.1 Mud accumulation sites

Mud flats form the archetypical example of coastal sites with high rates of sulfate reduction, and hence, substantial amounts of free sulfide are released into the porewater. This makes them a favourable habitat for cable bacteria. Previously, the presence





and activity of cable bacteria was already demonstrated in a subtidal mud flat off the Belgian coast (Malkin et al., 2014; van de Velde et al., in review). This site has permanently oxygenated overlying water and low levels of bioturbation, thus fitting the typical "e-SOx" site of Malkin et al. (2014) year round. In this study, we show that cable bacteria are also active in intertidal mud flats (see Table 1; Mokbaai and Cocksdorp), where oxygen is supplied from the air through exposure of the sediment at low tide. The observation that cable bacteria are potentially active within the deeper basin of Kobbefjord (Fig 6; Table 1) also fits the idea of cable bacteria being widely present in cohesive sediments. Fjords like Kobbefjord typically show an accumulation of fine-grained sediments, which is due to focusing of glacially derived material in the deeper basins of the fjords (Sørensen et al., 2015).

### 4.1.2 Bivalve reefs

Recently, it has been shown that cable bacteria proliferate in organic rich sediments that accumulate within mussel and oyster reefs in the Wadden Sea (Malkin et al., in review). Here, we present field observations of cable bacteria activity at two other bivalve reef sites and one cockle bed in the Wadden Sea (Eastern Scheldt and Mokbaai, Table 1 and Fig. 4), which suggests that cable bacteria are indeed prominently present in these bivalve reefs and beds. By increasing bed roughness and reducing local current velocity, as well as by depositing large quantities of fine particles as pseudofaeces, bivalve reefs show strongly enhanced accumulation of sediment, which is enriched in organics and fine particles (Bergfeld, 1999; van der Zee et al., 2012). Malkin et al. (in review) advanced an intriguing two-way ecological interaction between the bivalves and the cable bacteria, which could be beneficial to both. Through biodeposition of organic rich sediment, the bivalves stimulate sulfate reduction, and hence the growth of cable bacteria by increasing the supply of sulfide. On the other hand, the cable bacteria remove the sulfide within the top layer of the sediment, and thus detoxify the sediment from free sulfide, which is toxic to the bivalves (and other fauna). A similar two-way interaction could also be occurring in the cockle bed (Mokbaai, Table 1). The cockles are actively filtering seston from the overlying water, and in this way, they also enrich the surface sediment with organic matter and stimulate local sulfide production through sulfate reduction (Widdows and Navarro, 2007). However, unlike the sediment patches within oyster and mussel reefs, which show little bioturbation (Malkin et al. in review), cockles are known to induce a fair amount of particle mixing (Ciutat et al., 2006). In this sense, the observation of cable bacteria activity in the cockle bed is remarkable and does not fit the "low bioturbation" criterion as proposed by Malkin et al. (2014) – see further discussion in paragraph 4.3.

### 4.1.3 Salt marshes, seagrass beds and mangroves

Cable bacteria seem to be widely present in salt marshes. The first report of cable bacteria activity under natural conditions included a salt marsh in the Rhine-Meuse-Scheldt delta of the Netherlands (Malkin et al. 2014). More specifically, Malkin et al. (2014) sampled a mud accumulating creek bed in the salt marsh. These creeks receive large amounts of plant debris, and this constant supply of fresh organic matter promotes sulfate reduction in the sediment, thus increasing the levels of free sulfide in the porewater. Recent laboratory induction experiments reveal cable bacteria populations in sediments from creek beds and





salt marsh ponds in Canada and the USA (Larsen et al., 2015; Rao et al., 2016a). This study confirms that cable bacteria are widely active in the unvegetated areas of salt marshes, with new observations of e-SOx in salt marshes of the Dutch Delta (Eastern Scheldt, 3 sites, Table 1 and Figure 4) and the Wadden Sea (Mokbaai, 1 site, Table 1).

Similar to salt marshes, sediments near seagrass beds and mangrove trees, receive large amounts of plant debris. Due to wave attenuation by the seagrass bed or the root system of the mangrove trees, plant debris and organic matter accumulates. Cable bacteria are present and active in both these habitats. Burdorf et al. (2016) demonstrated *in situ* activity of cable bacteria in a temperate mangrove in Australia, while rRNA archives from Liang et al. (2006) point towards the presence of cable bacteria in tropical mangroves in China. In this study, we provide the first report of cable bacteria activity in seagrass beds: a laboratory enrichment of the sediment from the fringe of a Mediterranean seagrass bed (Étang d'Urbino, Corsica) developed an active cable bacteria population (Table 1, Fig. 7). Cable bacteria could potentially play an important role in minimizing the sulfide concentrations which can harm the rhizome system of seagrasses. Although most marine plants have a defensive mechanism in their root system (transporting oxygen through the roots to oxidize sulfide to the harmless sulfate), under certain conditions, the production of sulfide can outpace these defences (van der Heide et al., 2012). Note however, that our observations are from the fringe or unvegetated patches within seagrass beds or mangroves as microsensor profiling near the root-system was avoided due to the high likelihood of sensor breakage. Hence, the question whether cable bacteria are active directly next to the root system of marine plants, where they might possibly improve the conditions in the sediments for these marine plants, remains open.

### 4.1.4 Seasonally hypoxic coastal systems

Seitaj et al. (2015) observed that the occurrence of cable bacteria exhibits a seasonal pattern in the sediments of the seasonally hypoxic Lake Grevelingen, a coastal reservoir in the Netherlands. A multi-year study at one site showed that cable bacteria are abundant in the sediment in winter and spring, while *Beggiatoacaea* become dominant in fall after a period of summer hypoxia. Our investigations here show cable bacteria activity is present throughout Lake Grevelingen in deeper sediments (> 12 m water depth), thus revealing that cable bacteria activity occurs across the whole basin (Table 1 and Fig. 4). When present at high densities, cable bacteria control sedimentary sulfur oxidation, and by imposing acidic conditions in the suboxic zone, they induce the dissolution of the iron sulfides (FeS) in this layer. The mobilized iron resulting from FeS dissolution diffuses upwards, and causes the formation of a layer of iron (hydr)oxides in the oxic zone (Seitaj et al., 2015). In summer, this large pool of iron oxides acts as a "firewall" against the release of sulfide from the sediment to the water column, thus avoiding or delaying a detrimental environmental condition referred to as 'euxinia'. Overall, cable bacteria were shown to strongly influence the seasonal biogeochemical cycling in Lake Grevelingen through the regulation of the phosphorus cycle (Sulu-Gambari et al., 2016a), manganese cycle (Sulu-Gambari et al., 2016b) and iron cycle (Seitaj et al., 2015).

Cable bacteria activity might have a similar impact on other seasonally hypoxic systems. Sayama (2011) reported cable bacteria activity in the seasonally hypoxic Tokyo Bay (Japan) and a pH profile from the Santa Monica basin (USA; Reimers et al., 1996), an anoxic basin with occasional oxygen inflows, also indicates the presence of e-SOx. Here, we



demonstrate the rapid development of the e-SOx signal (e.g. less than 7 days) in another seasonally hypoxic system, the Yarra River estuary (Australia). Together these data indicate that cable bacteria seem to be particularly prevalent in these environments (Table 1 and Fig. 6).

5       This widespread presence of cable bacteria in seasonally hypoxic systems is not surprising since perturbation induced by seasonal oxygen depletion resembles that of the sediment incubation technique that is used to grow cable bacteria in the laboratory. In these sediment incubations, the infauna is removed from the system and sulfide is allowed to accumulate in the porewater, either by asphyxiation or by thoroughly mixing the sediment (hence mixing oxygenated sediment layers downwards into the bulk anoxic sediment). Such a "reset" of the sediment biogeochemistry apparently favours cable bacteria growth. The seasonal development of hypoxia in coastal bottom waters induces a similar reset of the sediment geochemistry: the hypoxia

kills most bioturbating fauna and stimulates benthic processes that produce sulfide. When the bottom waters become oxygenated again, the sediment depth profiles of oxygen and sulfide overlap, which provides ideal starting conditions for the growth of a cable bacteria (Schauer et al., 2014). The subsequent development of e-SOx can go fast. In sediments of the Yarra River site, the distinctive e-SOx pH signal developed in less than a week after the sediment was exposed to oxygen. In seasonally hypoxic systems, a periodic hypoxic event kills off most bioturbating fauna and stimulates anoxic benthic processes

that produce sulfide. The occurrence of coastal hypoxia is on the rise due to an increased anthropogenic input of nutrients into the coastal zone in combination with climate change (Diaz and Rosenberg, 2008). An improved understanding of the distribution and seasonality of cable bacteria in relation to seasonal hypoxia is hence needed.

### 4.1.5 Deep Sea Environments

Coastal sites have received a large part of the sampling effort in the search for cable bacteria, as they provide relatively easy

access for sampling. Dedicated studies on cable bacteria activity in deeper environments are still largely lacking. Gene archives, however, pinpoint to the presence of cable bacteria in the deep sea: 16S rRNA sequences similar to cable bacteria have been collected from sediments in the Nile Deep-Sea Fan (Grünke et al., 2011) and at a cold seep off New Zealand (Baco et al., 2010). Geochemical data also hint at the potential activity of cable bacteria in deeper environments, as field recordings of $O_2$ and $H_2S$ in sediments from the Mid-Atlantic Ridge imply potential cable bacteria activity at >3000 m depth (Schauer et

al., 2011). Finally, observations of "large filamentous bacteria" belonging to the *Desulfobulbacaea* group have been reported on the surface of inactive sulfide chimneys in the Southern Mariana Trough at 2810 m depth (Kato and Yamagishi, 2016), but it has yet to be determined whether these bacteria also display electrogenic activity. Overall, the distribution of cable bacteria within deep sea sediments is still largely unknown. Yet, given the reduced organic matter input, the deep oxygen penetration of the surface sediments in the abyssal plains and the absence of sulfate reduction, the vast ambient sediments of the deep sea

seem an unlikely habitat for cable bacteria. Based on the scarce evidence that is currently available, it appears that cable bacteria are restricted to localized hotspots, where sulfide is in sufficient supply, such as deep sea fans, hot vents and cold seeps.





### 4.1.6 Non-Marine Environments

Recent studies show that e-SOx is not limited to the marine realm, but that cable bacteria can also be active in freshwater environments. The presence and activity of cable bacteria was recently demonstrated in the surface sediment of a freshwater stream in Denmark (Risgaard-Petersen et al., 2015), and preliminary evidence suggests that cable bacteria might also be present
and active within the plume fringe of a hydrocarbon-contaminated aquifer in Germany (Müller et al., 2016).

### 4.2 Environmental controls on e-SOx

Given the observed occurrence of cable bacteria activity as discussed above, the question then rises as what are the constraints on the "ecological niche" of cable bacteria. In the next sections, we systematically discuss what is known about the
environmental controls on the distribution of e-SOx in the seafloor.

### 4.2.1 Electron donor availability

Microorganisms need suitable electron donors to ensure their energy supply that sustains their metabolism, and therefore, sulfide availability could be a key factor. A majority of the reported sites with abundant cable bacteria activity are indeed situated in productive areas with a high input of organic matter, which sustain high sulfate reduction rates, and hence high
rates of sulfide production (like salt marshes, bivalve reefs, mangroves and seasonally hypoxic basins). The specific pattern of cable bacteria activity that we observed within the Rhine-Meuse-Scheldt delta further supports the idea that sulfide availability could be a crucial constraint. The Rhine-Meuse-Scheldt is a former interconnected estuary, where the construction of dams has led to large differences in the biogeochemistry between the arms of the estuary. The most southern arm (Western Scheldt) has remained a true estuary with freshwater inflow, while the basin just north (Eastern Scheldt) has been cut off from riverine
input. The riverine input of iron (derived from terrestrial weathering) combined with high intensity of bioturbation favours dissimilatory iron production in the Western Scheldt. Accordingly, cohesive sediments in the Western Scheldt are typically more ferruginous compared to muddy deposits within the Eastern Scheldt, which tend to have higher rates of sulfate reduction. This geochemical difference is reflected in the presence of cable bacteria. Our survey in November 2015 showed the presence of e-SOx in >80% of the sampled (muddy) sediments of the Eastern Scheldt and no presence of e-SOx in the sampled (muddy)
sediments of the Western Scheldt (data not shown). Even though this sampling does not completely exclude the presence of cable bacteria in the Western Scheldt, it does show that cable bacteria thrive better in the sulfide-rich sediments of the Eastern Scheldt. Note that high sulfide availability does not necessarily required high $\Sigma H_2S$ concentrations in the porewater (i.e., one needs a high production rate, not a high stock). Cable bacteria have been observed to thrive in sediments with low concentrations of free sulfide in the porewater (<5 µM in a freshwater stream, Denmark (Risgaard-Petersen et al., 2015), and
<10 µM at a coastal site, St130, Belgium (van de Velde et al., in review). In those sites a cryptic sulfur cycle is thought to be active: sulfide is produced through sulfate reduction and iron-sulfide dissolution (enhanced by the acidification of the suboxic



zone by e-SOx), and this sulfide is immediately consumed through e-SOx, which then leads to a low concentration of free sulfide in the porewater.

### 4.2.2 Electron acceptor availability

Microorganisms also need access to electron acceptors to sustain their respiratory metabolism. Cable bacteria use either oxygen or nitrate as the terminal electron acceptor (Marzocchi et al., 2014). The availability of $O_2$ was high (>70% air saturation in the overlying water) at all the field sites where cable bacteria activity was detected in this study. Furthermore, in seasonally hypoxic sites, the cable bacteria are present in winter and spring when the overlying water is oxygenated, and they disappear in summer when hypoxia develops (Seitaj et al., 2015), thus confirming the importance of oxygen availability. The exact threshold level of $O_2$ in the overlying water, above which cable bacteria can survive and grow, is currently, however, not known and needs further study. Nevertheless, the presence of cable bacteria in intertidal areas does indicate that cable bacteria are capable of dealing with varying oxygen pressures. Diurnal fluctuations due to photosynthesis by microphytobenthos can drastically change the $O_2$ concentrations in the top layer of the sediment, thus leading to a dynamic repositioning of the OPD. To retain access to $O_2$, this requires that cable bacterial filaments follow the rhythms that are imposed on the oxygen availability in the sediment. A study by Malkin and Meysman (2015) showed that cable bacteria quickly responded to a reduction of the OPD when sediment was placed in the dark and photosynthesis by microphytobenthos ceased. To explain this response, the authors hypothesized that cable bacteria are able to migrate towards the oxygen and re-orient themselves in the sediment. Recent microscopical observations of sediments confirm this migration hypothesis (Bjerg et al., 2016).

### 4.2.3 Temperature and salinity conditions

Cable bacteria apparently can tolerate a wide range of temperature conditions. Their geographical distribution does not seem to be limited by temperature: a 0°C incubation in Greenland and a 25°C incubation in Australia both resulted in the development of an active cable bacteria population. Furthermore, the field observation of cable bacteria in mangrove sediment in Southern Australia in the summer suggests that cable bacteria can thrive at even higher temperatures. An even wider tolerance range is observed with respect to the salinity. e-SOx has been reported in freshwater sediments (Risgaard-Petersen et al., 2015), brackish sediment (Yarra River, Australia; this study), marine sediments (the majority of field observations up to present) and a hypersaline environment (Sabkhet Arina salt pan, Tunisia, salinity 40; this study).

### 4.3 Biotic controls on cable bacteria distribution: bioturbation

Bioturbation refers to the reworking of the surface sediment by infauna (Meysman et al., 2006). Bioturbating fauna have been hypothesized to be a main controlling factor in the occurrence of cable bacteria (i.e. Pfeffer et al. 2012, Malkin et al. 2014). As cable bacteria filaments span the sediment over several centimetres, this network could be mechanically damaged by burrowing organisms, thus leading to the collapse of the e-SOx activity. The laboratory mimic of this process, performed by pulling a thin wire through the suboxic zone, indeed leads to an immediate collapse of the distinctive pH maximum in the top



layer of the sediment (Pfeffer et al. 2012, Vasquez-Cardenas et al. 2015). Bioturbation also stimulates the cycling of iron in the sediment, thus favouring dissimilatory iron reduction over sulfate reduction (Canfield et al., 1993). Hence, bioturbation could impede cable bacteria development in a second way by reducing the production rate of sulfide. Because of this, it has been previously suggested that bioturbation exerts restrictive controls on the natural distribution of e-SOx in the seafloor (Malkin et al., 2014).

However, this idea may need some reconsideration. Recent field observations of cable bacteria in bioturbated mangrove areas (Burdorf et al. 2016) and cockle beds (this study), suggest that e-SOx and bioturbation are not mutually exclusive. To more closely examine the relationship between bioturbation and cable bacteria activity, we installed six mesocosms just outside our laboratory facility (NIOZ Yerseke, The Netherlands). Each mesocosm (l:0.92m x w:1.12m x h:0.63m, volume: 650 L) was filled ¾ with defaunated sediment retrieved from a bioturbated sand flat within the Eastern Scheldt (Oude Bieten Haven: 51°26'52"N; 04°05'47"E). To half of the mesocosms, a population of lugworms (*Arenicola marina*) was added (~50-60 individuals m$^{-2}$), while the other mesocosms remained devoid of large infauna. Triplicate cores were sampled from each mesocosm in August 2014 and cable bacteria activity was detected in all six mesocosms (Fig. 9). The depth of the pH minimum was located at approximately the same depth (+/- 15 mm) in both treatments, which indicates that e-SOx was present in the top 15 mm of the sediment. Yet, the distinctive pH excursions were larger in the non-bioturbated mesocosms compared to the bioturbated treatment (Fig. 9). The suboxic zone in the bioturbated tanks was significantly larger than in the non-bioturbated tanks, which was probably the result of the ventilation of the porewater by *A. marina*. These results confirm that cable bacteria are active in bioturbated sediments.

Bioturbation is a concept that covers a diverse group of infauna, which display a variety of sediment reworking activities (e.g. burrowing, moving, feeding). Accordingly, in a bioturbated sediment, different patches will be disturbed with a different frequency and intensity. This heterogeneity with which the sediment is reworked, combined with the fast growth of cable bacteria, can be the key to the observed co-existence of cable bacteria and bioturbating fauna. Laboratory induction experiments (e.g. Fig. 6) have shown that cable bacteria can build up a sizeable activity in less than a week's time. In a bioturbated habitat, if a patch of sediment is not reworked within one week, cable bacteria would be able to develop. A high bioturbation pressure could thus induce a frequency of patch reworking too high for cable bacteria to develop. But a medium "disturbance" pressure from bioturbation might even be beneficial to the bacterial growth, as the input of fresh organic material (due to filter feeding fauna) can stimulate sulfate reduction and thus replenish the sulfide pool available to cable bacteria.

## 5 Conclusion

Cable bacteria are globally present and active in marine sediments. Within coastal areas, cable bacteria are active in a variety of habitats (mud flats, seagrass beds, mangroves, salt marshes, seasonally hypoxic basins) and across all latitudes (from tropical to polar environments). Sampling in a restricted geographical area (The Netherlands) revealed that cable bacteria are found in many locations and at different time points in these coastal habitats. The high ratio of successful detection over the sampling





effort suggests a widespread occurrence. We contend that similar sampling campaigns in other geographical locations would reveal a similar common occurrence of cable bacteria activity in coastal habitats. Previous studies have documented a considerable impact of e-SOx on the elemental cycling (Risgaard-Petersen et al., 2012) and fluxes across the sediment-water interface (Rao et al., 2016). The combination of a widespread occurrence and a strong local geochemical imprint suggests that

cable bacteria could be important in the cycling of carbon, sulfur and other elements in coastal environments. In contrast, the deep sea remains largely unexplored with respect to the presence and activity of cable bacteria. Based on the scarce evidence that is currently available, cable bacteria seem to be restricted to localized hotspots where sulfide is in sufficient supply (deep sea fans, hot vents, cold seeps).

This study also provides a better insight to the environmental constraints on the natural distribution of cable bacteria within

the seafloor. Cable bacteria are capable of growing in a wide range of temperature and salinity conditions. The ecological niche of cable bacteria appears to be primarily constrained by the availability of suitable electron acceptors ($O_2$/$NO_3$) and the availability of free sulfide as the electron donor (liberated in the porewater through sulfate reduction or through the dissolution of FeS). Sediment disturbance through bioturbation appears to mechanically impede the filament network of cable bacteria, but fast regrowth occurs when a sediment patch is left undisturbed for a sufficiently long period. The observation that cable

bacteria are also present in bioturbated sediments greatly extends their potential distribution in the present-day seafloor (where most sediments are bioturbated).

### Acknowledgements

We thank all collaborators who hosted and facilitated the numerous field sampling campaigns. For Long Island Sound we

thank R.C. Aller, Q. Zhu, J. Soto-Neira, and C. Heilbrun and the crew of the R/V Sea Wolf for their support during sample collection. We are also grateful to J. Soto-Neira, and C. Heilbrun for the help during the laboratory incubations. For the Australian campaigns, we thank P.L.M. Cook, E.K. Robertson and V. Eate for the help and advice during sampling and incubation. For the sampling on Texel, we thank S. Nieuwhof, H. de Stigter and the effort of students (M. Los, D. van Loon, I. Klarenberg and C. van der Weijst). For the sampling on the Lake Grevelingen we thank the crew of the R.V. Navicula. For

sampling in Corsica, we thank the Corbara holiday team, while Dr. W. Oueslati helpfully assisted in the Tunisian field work. This research was financially supported by the European Research Foundation (ERC Grant 306933 to FJRM), and the Flanders Research foundation (FWO aspirant grant to L. Meire), the Department for Education, Church, Culture and Equality (IIKNN Greenland) and the Schure-Beijerinck-Popping Fonds (SBP2013/59).18 to DS), and the Darwin Center for Biogeosciences (DS).



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



**Tables**

Table 1: Summary of the globally samples sites where cable bacteria activity was measured or presence of cable bacteria was demonstrated. Full details of all locations are given in Table S1.

| Site name | Sampling year(s) | N° locations | Country | Habitat type | Activity evidence Field | Activity evidence Lab | Presence evidence Field | Presence evidence Lab |
|---|---|---|---|---|---|---|---|---|
| **Grevelingen** | 2012 - 2015 | 9 | Netherlands | Seasonally hypoxic lake | X | X | X | X |
| **Kobbefjord** | 2011 | 1 | Greenland | Fjord sediment | | X | | |
| **Yarra River** | 2014 | 1 | Australia | Seasonally hypoxic estuary | X | X | X | X |
| **Long Island Sound** | 2013 | 1 | United States | Seasonally hypoxic basin | | X | | X |
| **Urbino lagoon** | 2015 | 1 | Corsica, France | Fringe of seagrass bed | | X | | X |
| **Sabkhet Arina** | 2015 | 1 | Tunisia | Salt flat | | X | | X |
| **Station 130** | 2015 | 1 | Belgium | Subtidal mud flat | X | X | X | X |
| **Mokbaai** | 2014 – 2015 | 5 | Netherlands | Oyster reef | X | X | | X |
| | | | | Salt marsh | X | X | | X |
| | | | | Cockle reef | X | | | |
| **Cocksdorp** | 2014 - 2015 | 3 | Netherlands | Oyster/Mussel reef | X | X | X | X |
| **Eastern Scheldt** | 2012 – 2016 | 8 | Netherlands | Salt marsh | X | X | X | X |
| | | | | Intertidal mud flats | X | X | X | X |
| | | | | Bioturbated sand flat | | X | | X |
| **Western Scheldt** | 2013 | 1 | Netherlands/ Belgium | Oxygenated estuary | X | | | |

**Figures**

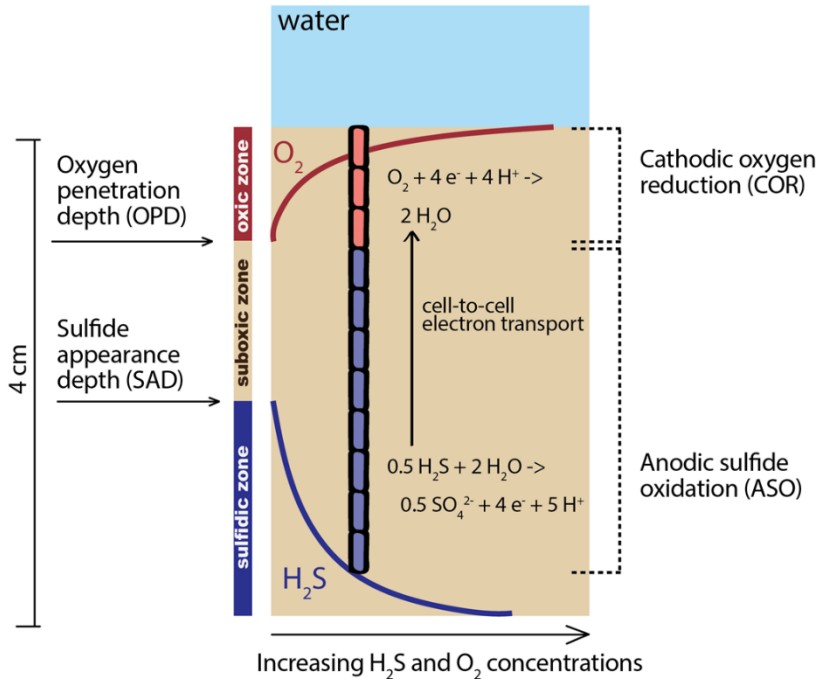

**Figure 1:** Schematic illustration of the metabolism of a cable bacteria. Electrogenic sulfur oxidation involves two spatially separated redox half reactions, which are referred to as the cathodic oxygen reduction (COR) and the anodic sulfide oxidation (ASO). The redox coupling is ensured by an electron transport axis through the cable bacteria filament. The metabolism of cable bacteria leads to three distinct geochemical zones in the sediment: the oxic zone delineated by the sediment-water interface and the oxygen penetration depth (OPD), the suboxic zone defined as the sediment zone where neither $O_2$ nor $H_2S$ is detectable, and the sulfidic zone, where free sulfide accumulates below the sulfide appearance depth (SAD).




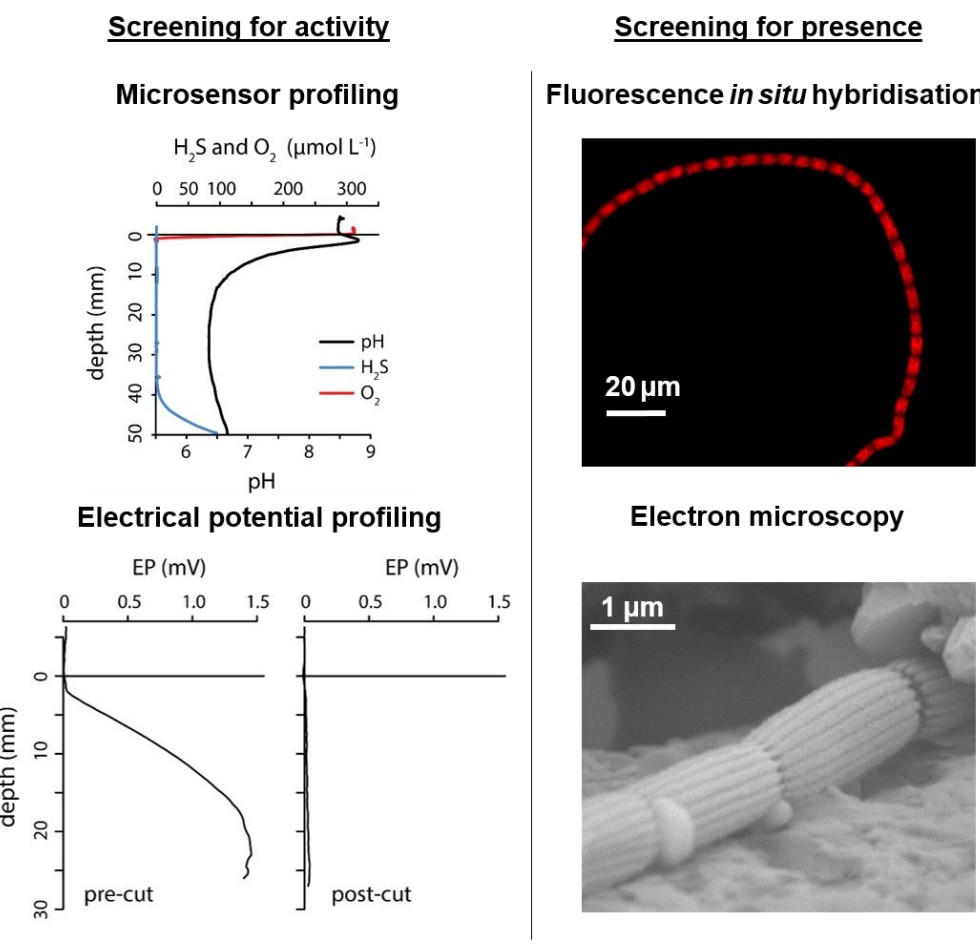

**Figure 2:** Overview of the methods that demonstrate cable bacteria activity (left column) and cable bacteria presence (right column) in aquatic sediments. Microsensor profiling produces high-resolution depth profiles of pH, $H_2S$ and $O_2$ which reveal the characteristic geochemical fingerprint of electrogenic sulfur oxidation (pH excursions and suboxic zone). Electrical potential profiling directly demonstrates the build-up of the electrical potential in the sediment due to long-distance electron transport (Damgaard et al., 2015; Risgaard-Petersen et al., 2015). Passing a thin wire through the sediment cuts the cable bacteria filaments, and confirms that the observed build-up of the electrical potential is due to cable bacteria activity. The presence of cable bacteria can be confirmed by RNA staining (FISH probe DSB706) or by electron microscopy, which reveals the parallel ridges on the cell envelope that are unique to cable bacteria.




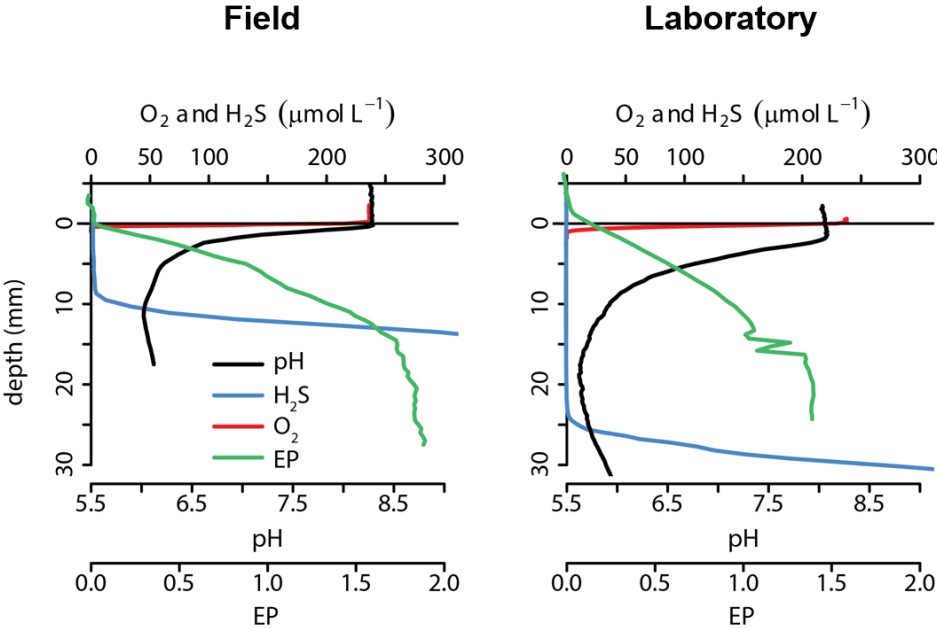

**Figure 3:** High-resolution depth profiles reveal the typical fingerprint of cable bacteria activity in sediments near a bivalve reef (Cocksdorp, The Netherlands). (Left panel) Field measurements on intact sediment cores. Profiles were recorded within 3 hours of core collection. (Right panel) Incubation of sediment from the same location under laboratory conditions. Sediment was sieved and homogenised before incubation. Profiles were recorded after 19 days.



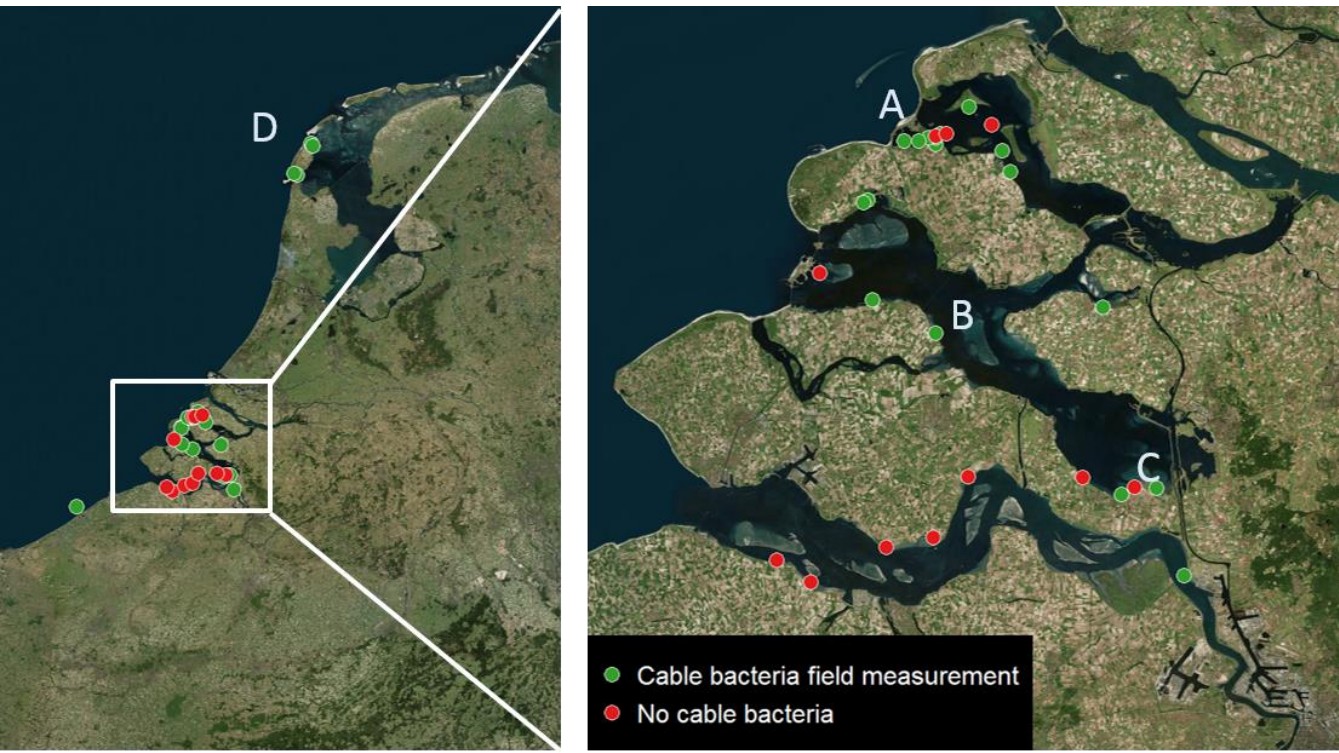

**Figure 4:** Field site locations examined for cable bacteria activity within a close range to the home laboratory (Belgium and the Netherlands). Green markers indicate locations where cable bacteria activity was found. Red markers indicate locations where cable bacteria activity was not detected. The letters A-D indicate the locations of the sites detailed in Figure 5.




| **Site** | **Activity** | **Presence** |
|---|---|---|

### A. Seasonally hypoxic lake

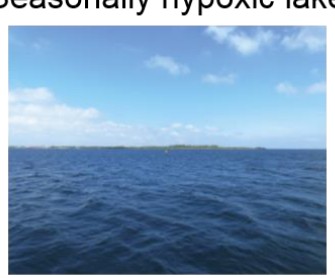
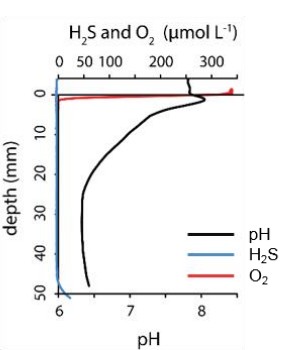
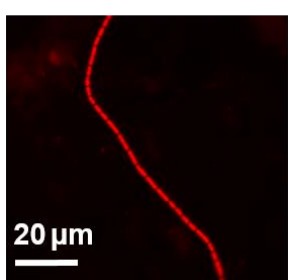

### B. Bivalve reef

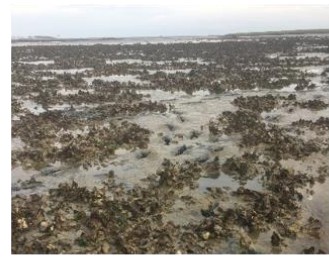

### C. Salt marsh

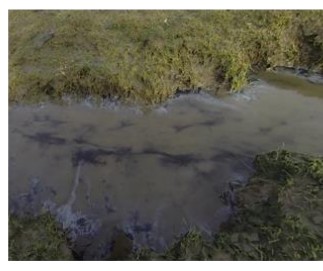
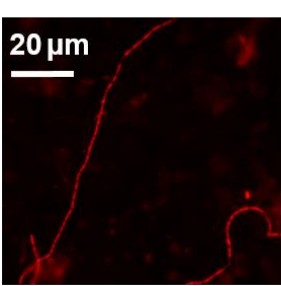

### D. Intertidal mud flat

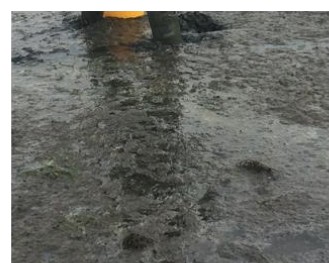
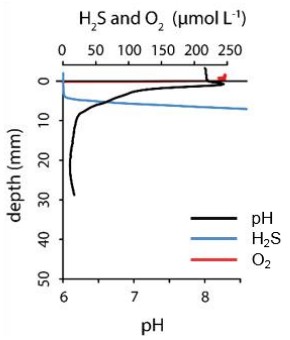
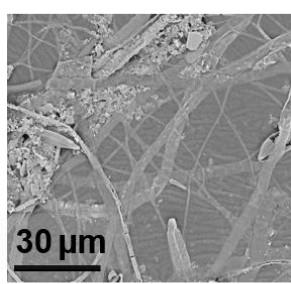



**Figure 5:** Results from four marine habitats in the Netherlands exemplifying detected cable bacteria activity and presence: (1) sediments in deeper (> 15 m) basins of the seasonally hypoxic Lake Grevelingen, (2) a bivalve reef in the Eastern Scheldt, (3) a salt marsh in the Eastern Scheldt (Rattekaai) and (4) an intertidal mud flat in the Wadden Sea (Mokbaai). Locations are marked by letters A–D in Fig. 4. Left column: photographs of the sampling locations. Middle column: microsensor profiles documenting the geochemical fingerprint of cable bacteria activity. Right column: the presence of cable bacteria was confirmed using microscopy techniques (either by fluorescence in situ hybridization (A,C) or scanning electron microscopy (B,D).





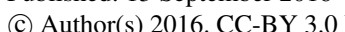

**Figure 6:** Microsensor depth profiles from laboratory incubation experiments within three widely different climatic regions. Sediments were profiled at regular time intervals during incubation. Top row: sediment from Long Island Sound (New York, USA) showed the characteristic pH excursions of e-SOx after 7 days of incubation. Middle row: sediment from the Yarra River (Melbourne, Australia) showed cable bacteria activity after 4 days of incubation. Bottom row: sediment from Kobbefjord (Nuuk, Greenland) showed the pH fingerprint of cable bacteria activity after 26 days of incubation.



## Corsica: Fringe of seagrass bed

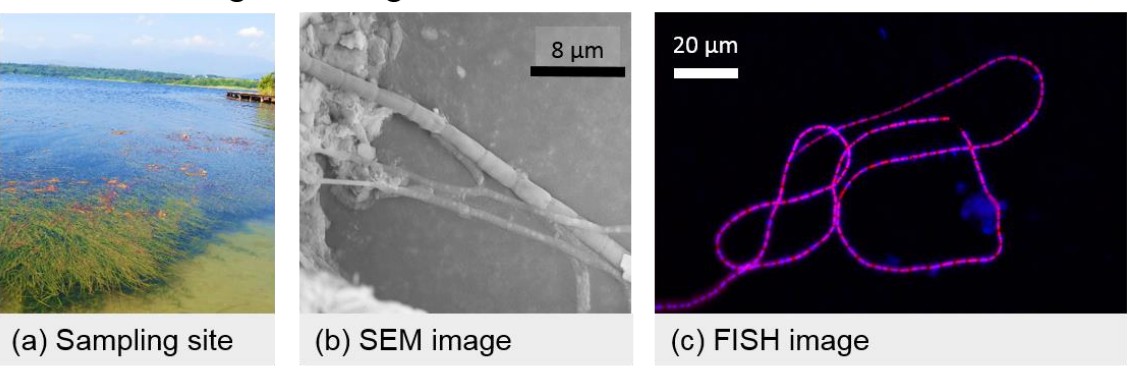

## Tunisia: Coastal salt pan

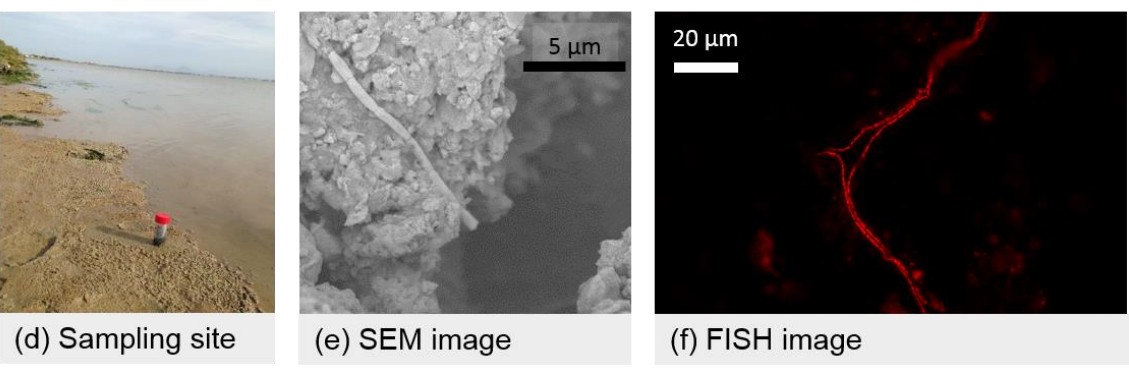

**Figure 7:** Microscopy reveals the presence of cable bacteria in two different habitats in the Mediterranean. (a-c) Sediment near a seagrass bed in the Urbino lagoon (Corsica, France) (d-f) Sediment near the water line of the salt pan Sabkhet Arina (Tunis, Tunisia). Left image: photograph from the sampling sites. Middle image: scanning electron micrographs of cable bacteria filaments with the typical ridge pattern. Right image: Fluorescence in situ hybridization (FISH) images using a DSB706 probe.



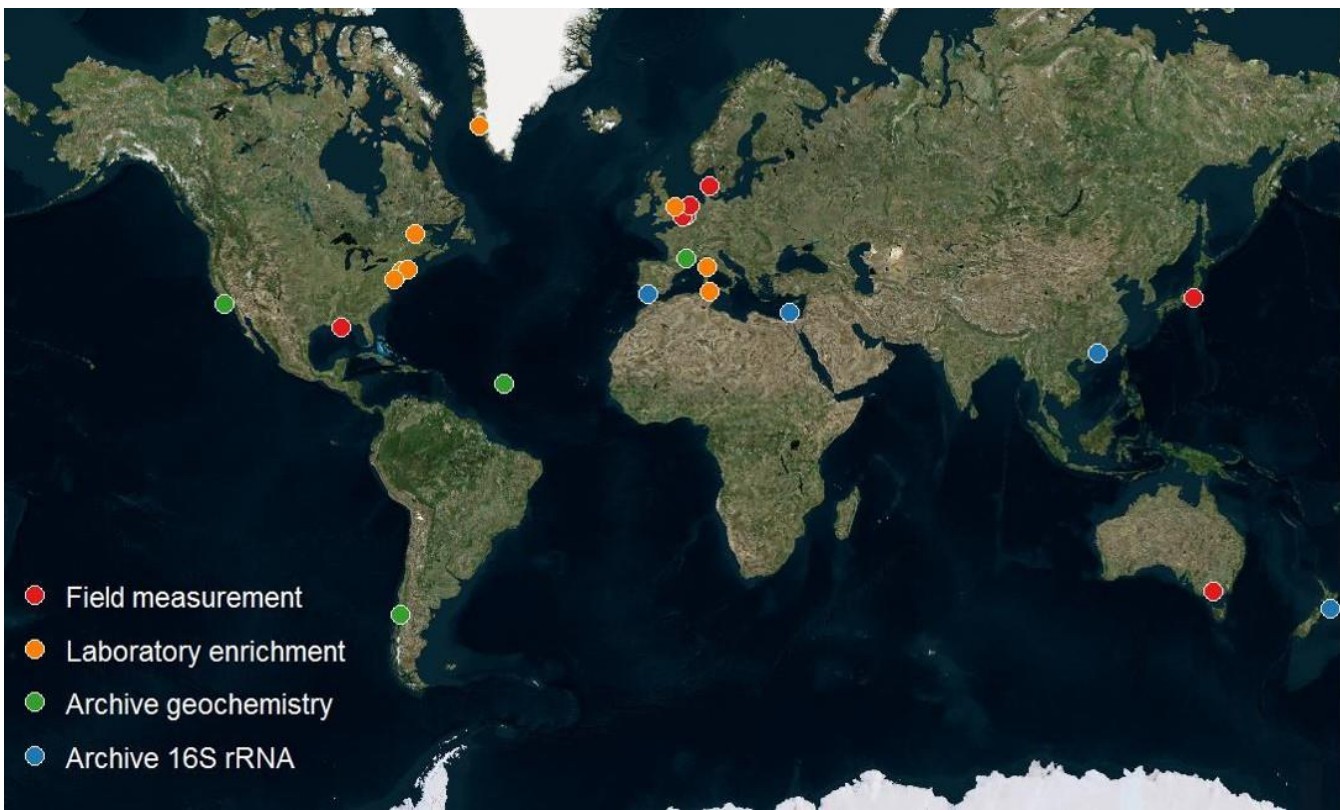

**Figure 8:** Global distribution of cable bacteria activity and/or presence in marine sediments. The colours of the markers represent the screening method that was used to detect cable bacteria presence or activity. Red markers: direct field observation of cable bacteria activity. Orange markers: cable bacteria activity demonstrated after a laboratory induction experiment. Green markers: published geochemical profiles with an e-SOx signature, but which were not interpreted as such at the time of publication. Blue markers: sites with reported 16S rRNA sequences that have a >97% similarity to cable bacteria.





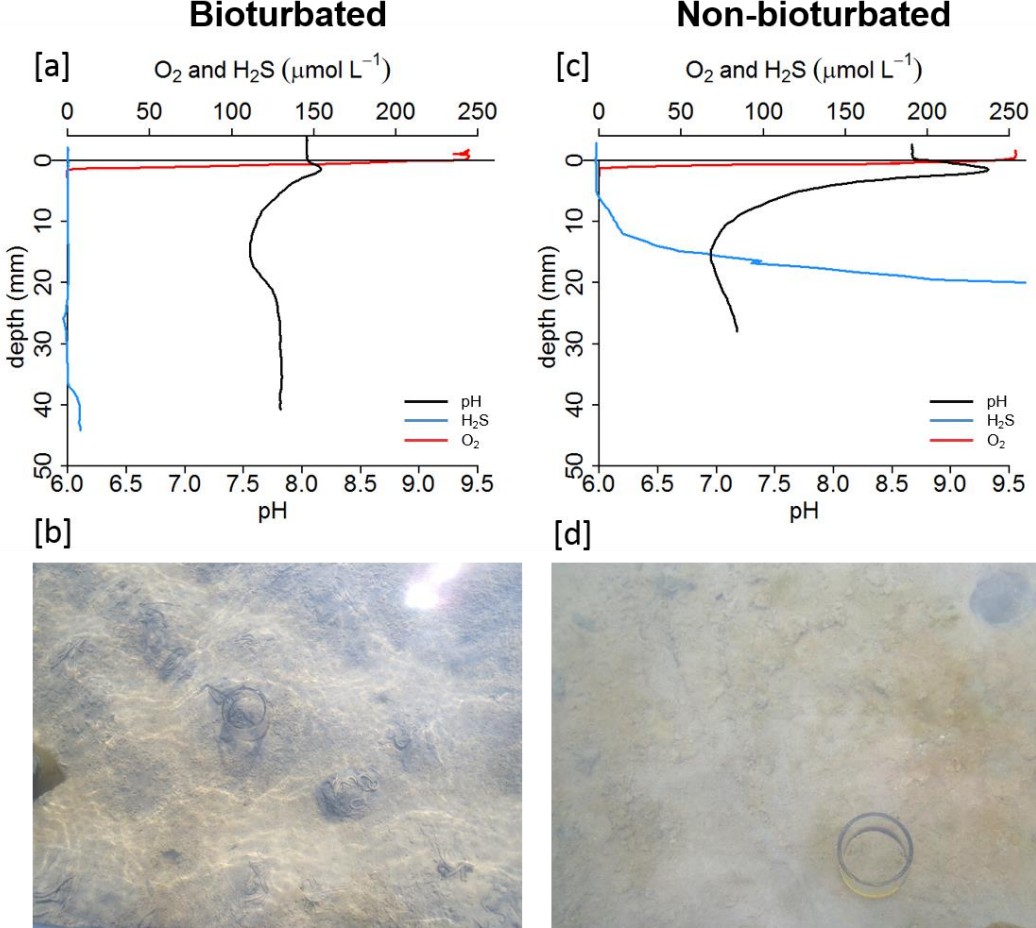

**Figure 9:** Activity of cable bacteria in mesocosm experiments with two treatments: bioturbated (sediment with the addition of lugworms) and non-bioturbated (control sediment without lugworm addition). [a] Microsensor depth profiles reveal an active cable bacteria population in the bioturbated mesocosm. [b] Surface appearance of the bioturbated sediment showing faecal casts of lugworms. The coring location for microsensor profiling is indicated with the core liner. [c] Geochemical fingerprint in non-bioturbated sediment shows strong cable bacteria activity. [d] Surface appearance of the non-bioturbated sediment with core location for microsensor profiling.