# Peer review of "Long-distance electron transport occurs globally in marine sediments"

_Biogeosciences, 2016_

## Referee Comment (RC1) · Anonymous Referee #1 · 29 Sep 2016

Burdorf and Co-workers describe and discuss the distribution of Cable bacteria and the associated e-sox in marine environments. In addition they provide a description of the current methodology used to access the bacteria and electrogenic sulfur oxidation process. Their description of cable and e-sox abundance is partly based on published data but also to a large extend on currently unpublished data collected by the authors. Although a similar survey is included in the Malkin 2014 paper the study of Burdorf et al. is justified by the systematic inclusion of data from a variety of marine environments which has not been screened for cable bacteria before. The study provides a good overview of the distribution of cable bacteria and e-sox in the marine environment and given the novelty of the discovery of the bacteria and the process such and overview is important. I therefore recommend publication.

---

## Referee Comment (RC2) · C. Reimers (Referee) · 19 Oct 2016

This article addresses the question of how widespread is the occurrence of electrogenic cable bacteria in marine sediments. These novel bacteria were discovered in recent years and found to produce unique biogeochemical signatures in surface sediments. Oxygen reduction and sulfide oxidation mediated over centimeter scales by cable bacteria are readily detected by measuring microsensor profiles of porewater dissolved oxygen, pH, hydrogen sulfide and electrical potential. Cable bacteria belong to the family Desulfobulbaceae and may be identified by fluorescence in situ hybridization molecular staining techniques and by electron microscopy. The paper describes how a combination of these methods has been applied to detect cable bacteria presence in sediments from locations from around the globe.

[Figure]

I read the paper with great interest and offer only a few comments/suggestions for consideration along with some minor editorial suggestions.

Comments:

1. A distinctive ridge pattern that runs in parallel to the longitudinal axis of cells in filaments is described by the authors in section 2.6 of the paper as a characteristic that "unequivocally confirms that cable bacteria are present". This pattern is observed using scanning electron microscopy and illustrated in Figure 2. However, other SEM images of filaments in Figures 5 and 7 do not appear to show longitudinal ridges. How do the authors explain this? Are there cable bacteria species that do not have the ridges? Can the ridges be covered by sheath materials? Do the authors attribute cell diameters of 0.7-3 $\mu$m to be all inclusive and/or to indicate different species?

2. The direct measurement of an electrical potential (EP) that is generated in the sediment in the presence of electrogenic sulfide oxidation separated in space from oxygen reduction is described as a means to demonstrate cable bacteria activity. This EP is likely a new concept for most readers, and its measurement involves a new microsensor described by Damgaard et al. (2014). The paper would be improved if more background information was given to explain the origin of these electrical potentials and how they differ from the concept of sediment "redox potential". What does it mean that a voltmeter was used for the measurements with "MB 11mV"? How much signal drift was observed? If as stated the authors averaged readings taken from two profiles, one going down into the sediment and the second in reverse, then measurements at the base of a profile would be measured close in time (with little drift) while measurements at the top of the profile would be affected by the most possible drift. This could impose drift artifacts. Furthermore, how was the salinity of the overlying water "set to match the salinity of the porewater"? How small a difference in salinity in the overlying water would bias the EP measurements?

3. Profiles of pH that show a near surface maximum and low subsurface values leading

to pH excursions > 1 pH unit are described as indicative of spatially separated redox half reactions mediated by cable bacteria. The authors note in section 4.1.4 a pH profile from Santa Monica Basin measured by Reimers et al. (1996) has this form. However, they say little more about this location or the profile. Several studies have characterized Santa Monica Basin porewater chemistry including a paper by Jahnke, R. A. (1990) Early diagenesis and recycling of biogenic debris at the seafloor, Santa Monica Basin, California. J. Mar. Res 48:413-436. Of note, bottom water pH values are relatively low in Santa Monica Basin and so is the bottom water dissolved oxygen concentration. This may be why the pH excursion observed is only about 0.4 pH units. It would seem that sites where the overlying water is in equilibrium with atmospheric oxygen and atmospheric $CO_2$ are more poised to produce the large pH excursions described by the authors.

Minor Editorial Suggestions:

Page 1, Line 20: The locales listed here in the abstract are not "oceanographic regions or climate zones". It would be more accurate to say "(coastal sites off The Netherlands, Greenland, the USA and Australia)".

Page 1, Line 29. Change to "induce electron transport" (drop "an").

Page 2, Line 4. Change to "electro-active marine sediments hosting cable bacteria."

Page 2, Lines 24-25. Change to "cable bacteria may extend well beyond the sediment and impact the water column chemistry".

Page 2, Lines 33-34. Change to "only a few reports from a restricted number of coastal sites and habitats describe cable bacteria."

Page 3, Line 13. Change to "cable bacteria in seafloor sediments". If you were considering "the seafloor" you would be considering all types of substrates including bare rock. Same comment applies on page 4, line 10.

Page 4, Line 18. Reword as "Metabolic activity in the field can be detected either by in

situ microsensor profiling. . ... either on shipboard. . .”

Page 4, Line 23. Reword as “Cores from deeper areas were retrieved using a gravity core sampler. . ..”

Page 5, Line 5. Reword as “in a fashion similar to previous studies. . .”

Page 5, Line 9. Reword as “allows assessment of cable bacteria effects on the. . ..”

Page 6, Line 18. Provide after “the R package marelac” some reference or description to explain what this is, who developed it, and where it is available.

Page 16, Line 13. Reword as: “The subsequent development of e-SOx can be rapid.”

Page 16, Line 21. The evidence here is only suggestive so best to reword as “however, suggest the presence of cable bacteria. . ..”

Figure 3. Give units for EP.

––––––––––––––––––––––––––––––

---

## Referee Comment (RC3) · Anonymous Referee #3 · 27 Oct 2016

The manuscript describes the world wide distribution of cable bacteria in marine sediments. It's a thorough work and interesting to the readers of the journal. Major comments: 1) A major concern is the style of the manuscript. It is written in a very lenghty, prosaic style with numerous repetitions between the materials, results, and discussion parts. The manuscript can be easily shortened to half. 2) The materials, results, and discussion are not well seperated but appear very mixed. Lots of materials are presented in the results etc. 3) The manuscript leaves the impression that LDET is the most dominant type of sulfur metabolism in marine sediments which might be true or not. It should be compared to the classical redox sequence model which has been found almost everywhere until recently and the overall importance of the two models should be discussed. 4) I suggest to not use the term eSOX. The electricity component is added by humans. The bacs don't generate electricity. The fact that humans can

measure an electric field under very high activities and high resistance of the sediment does not mean that an electric field is always generated in nature. It's a very catchy term but not scientifically correct in my eyes. 5) The data can be presented in a more systematic way like in table 1. I would not present all the detailed figures which all show the same content. This would also contribute to a drastic shortening of the manuscript.

Specific comments: 6) Page 5, line 17-30: repetition 7) P6, l 8-11: repetition 8) P6, l 29-P7, l4: should be in discussion 9) P 7, l 8: what is modified precisely? 10) P8, l 12-17: repetition 11) P8, l 24-31: should be in discussion 12) P9, l 1-9: should be in Materials and is a repetition 13) P9, l 11: I suggest to need abbreviate OPD. Makes text more difficult to read. 14) L 12: what delta pH? Delta between what values? 15) L 14: replace oxygen penetration depth by oxic zone 16) L 15-22: This comparison cannot be done because the gradients and the zones change over time. Even for lab incubations such a comparison would only be feasible if the profiles are measured at distinct times and if it were shown before that the generation of the profile at a certain time is reproducibly and predictable. However, these gradients are dynamic and even collapse after a certain time. The situation is even worse in the field, as the kinetics of the profile formation are not known and the time point of sampling within such a built up or collapse time period is not defined. 17) L 24-30: Materials 18) P9, l 2-14: Materials 19) L 28: repetition 20) L 32-33: delete sentense 21) From here on I stopped indicating that parts belong to Materials or other sections 22) P13, l 9: crucial player: what is crucial? Any estimates of the quantitative importance? 23) L 10: marine sediments: is that only coastal or all? 24) L 8-19: repetition 25) L 23: fine-grained: data on sediment properties lacking in Materials part. A characterization of the sediments of all sites would be good. 26) P 19, l 6-18: This is obviously a new experiment which should be described in materials and in results accordingly. 27) P 19: Conclusion is more a second abstract than a conclusion. 28) Figure 1: unnessassary abbreviations are introduced which are not used at any other place in the manuscript. 29) Figure 5: name sites (1-4) the same as the panels (A-D) 30) Fig. 5: Indications of presense seems arbitrary. I suggest presenting the data in a table to make it comparable. Electron micrographs do not

show ridges here! 31) Fig. 9: New experiment –> materials? Results?

---

## Author Comment (AC1) · 17 Nov 2016

**Reviewer 1:**

Burdorf and Co-workers describe and discuss the distribution of Cable bacteria and the associated e-sox in marine environments. In addition they provide a description of the current methodology used to access the bacteria and electrogenic sulfur oxidation process. Their description of cable and e-sox abundance is partly based on published data but also to a large extend on currently unpublished data collected by the authors. Although a similar survey is included in the Malkin 2014 paper the study of Burdorf et al. is justified by the systematic inclusion of data from a variety of marine environments which has not been screened for cable bacteria before. The study provides a good overview of the distribution of cable bacteria and e-sox in the marine environment and given the novelty of the discovery of the bacteria and the process such and overview is important. I therefore recommend publication.

**Response**

We thank the reviewer for these encouraging, kind words.

**Reviewer 2:**

This article addresses the question of how widespread is the occurrence of electrogenic cable bacteria in marine sediments. These novel bacteria were discovered in recent years and found to produce unique biogeochemical signatures in surface sediments. Oxygen reduction and sulfide oxidation mediated over centimeter scales by cable bacteria are readily detected by measuring microsensor profiles of porewater dissolved oxygen, pH, hydrogen sulfide and electrical potential. Cable bacteria belong to the family Desulfobulbaceae and may be identified by fluorescence in situ hybridization molecular staining techniques and by electron microscopy. The paper describes how a combination of these methods has been applied to detect cable bacteria presence in sediments from locations from around the globe.

I read the paper with great interest and offer only a few comments/suggestions for consideration along with some minor editorial suggestions.

We thank Clare Reimers for this constructive and thoughtful review.

Comments:

1. A distinctive ridge pattern that runs in parallel to the longitudinal axis of cells in filaments is described by the authors in section 2.6 of the paper as a characteristic that "unequivocally confirms that cable bacteria are present". This pattern is observed using scanning electron microscopy and illustrated in Figure 2. However, other SEM images of filaments in Figures 5 and 7 do not appear to show longitudinal ridges. How do the authors explain this? Are there cable bacteria species that do not have the ridges? Can the ridges be covered by sheath materials? Do the authors attribute cell diameters of 0.7-3 _m to be all inclusive and/or to indicate different species?

**Response to comment 1**

We thank the reviewer for this justified remark. This same comment is also brought up by reviewer 3. The SEM-images provided in the manuscript were small and of low magnification, and so the longitudinal ridges of the cable bacteria were not visible. We now provide clearer

images that zoom in onto the cable bacteria, so that the longitudinal ridges are clearly visible . This has been done for Figures 5d, 7b and 7e. We have additionally included the associated full size (overview) images of these zoomed-in images in the supplemental information.

Until now, all cable bacteria detected by FISH probes do also display these characteristic longitudinal ridges. However, sometimes this outer topography is not visible, because the filaments are encrusted with minerals which masks the ridge pattern. This filament encrustation is especially notable in the (oxic) top part of the sediment, where most likely FeOOH-type deposition occurs on the outside of the cable bacteria.

The cell diameters of cable bacteria have indeed been shown to vary greatly between environments and also within a same sediment sample. We have observed in our lab in cell diameters ranging from 0.3 – 6 µm of cable bacterial cells. "Cable bacteria" is the name of a functional group of microbes and encompasses different species (Trojan et al. 2016). It is still unclear if cell width is the result of genotypic differences between the bacteria or whether it is a phenotypic reflection of the environment or the growth stage of the cable bacteria. The development of new species-specific FISH probes will hopefully allow us to further answer this question in the future.

**Changes in the text in response to comment 1**

(a) Replaced images

Fig 5.d. has been replaced with a SEM image of cable bacteria from the same sample with more distinctive ridges

Fig 7.b. has been further magnified in the image of the manuscript

Fig7.e. has been magnified in the image of the manuscript

Additionally full-size images of all SEM images are included in the supplemental material images S1 – S4

(b) Added text

(1)  P1: L30 – L32. "The cable bacteria form a mono-phyletic group within the *Desulfobulbacaea* family that currently encompasses six species of filamentous bacteria belonging to two genera "*Candidatus* Electrothrix" and "*Candidatus* Electronema" (Trojan et al. 2016)."

(2)  P7: L5 – L7. Until now, all cable bacteria detected by FISH probes (see section 2.7) also display these characteristic longitudinal ridges, and so the ridge pattern appears unique to cable bacteria (Fig. 2). Deposition of minerals onto the surface of the cable bacteria can however sometimes mask the ridge patterns. Such mineral encrustation is frequently observed in the oxic zone of the sediment, likely resulting from FeOOH deposition through ferrous iron oxidation.

2. The direct measurement of an electrical potential (EP) that is generated in the sediment in the presence of electrogenic sulfide oxidation separated in space from oxygen reduction is described as a means to demonstrate cable bacteria activity. This EP is likely a new concept for most readers, and its measurement involves a new microsensor described by Damgaard et al. (2014). The paper would be improved if more background information was given to explain the origin of these electrical potentials and how they differ from the concept of sediment "redox potential". What does it mean that a voltmeter was used for the measurements with "MB 11mV"? How much signal drift was observed? If as stated the authors averaged readings taken from two profiles, one going down into the sediment and the second in reverse, then measurements at the base of a profile would be measured close in time (with little drift) while measurements at the top of the profile would be affected by the most possible drift. This could impose drift artifacts. Furthermore, how was the salinity of the overlying water "set to match the salinity of the porewater"? How small a difference in salinity in the overlying water would bias the EP measurements?

**Response to comment 2**

We thank the reviewer for this remark. We agree that the EP-sensor is a novel technique for the most readers, and so we have now added some sentences in the materials section 2.5 to briefly explain the technique and to guide the readers to the appropriate literature for more detailed explanation (3).

"MB 11mV" is the model of our voltmeter used in the measurement for the EP-signal. We have modified the sentence to clarify this point (4).

The signal drift was estimated to be around +/- 0.06 mV per run (between bottom and top sediment). To avoid overestimating or underestimating the EP built-up in the sediment, two runs were measured. Because EP does not measure absolute values, but rather a current difference between the top and the bottom of the sediment, having opposite runs filters out the drift (assuming the drift is constant over time).

The bias imposed by salinity can be derived from a modified Nernst-equation, Planck-Henderson equation for diffusion (Damgaard et al. 2014 based on Bockris And Reddy 1998).

$$-\Delta\psi = \frac{RT}{F}\sum_i \frac{t_i}{Z_i}\ln\frac{C_i(l)}{C_0(0)}$$

where R is the universal gas constant, T is the temperature in Kelvin, F is the number of Faraday, $t_i$ is the transport number (diffusion constant) and $Z_i$ the valence of the ion i. Damgaard et al. 2014 calculate that a 10% change in salinity leads to a 0.55 mV diffusion potential.

For the laboratory incubations, the salinity was set and controlled done by preparing one large batch of artificial sea water (ASW). One part of this ASW was used to sieve the sediment before incubation and the other part of the sea water was used for the incubation of the cores. To verify that no difference in salinity was present we measured $Cl^-$ in the overlying water and the porewater.

For the field samples, water at the field site was collected and was used as overlying water during the EP measurement. After the EP-measurement, the top 5 cm slice of the sediment was centrifuged and the salinity was compared to that of the overlying water.

**Changes in the text in response to comment 2**

(3) "Cable bacteria impose an electron transport from the anodic cells in the anoxic zone to cathodic cells in the oxic zone (Fig. 2), which results in an ionic countercurrent of equal magnitude in the pore water. Due to a difference in mobility between negative and positive ions in the porewater, this ionic countercurrent creates a measurable difference in electrical potential (EP) between the surface and deeper sediment (Revil et al. 2010), which is in the range of 0.5-2 mV in marine sediments (Risgaard-Petersen et al. 2014, 2015). The electrical potential should not be confused with the redox potential, which expresses the tendency of a solution to take up or release electrons by redox reactions (Damgaard et al. 2014). While a gradient in redox potential is present in all coastal sediments and reflects the down-core variation in the oxidation state of the pore water, an EP gradient is only present in sediments showing either salinity gradients (potentials induced by ion diffusion) or sediment showing cable activity (potentials induced by long-distance electron transport)."

(4) Here, we employed EP microsensors built according to (and provided by) Damgaard et al. (2014). These custom-built EP sensors were connected in conjunction with a standard reference electrode (Radiometer, Denmark) to a of high impedance voltmeter (instrument type "MB 11mV", Microscale Measurement, The Hague, The Netherlands).

3. Profiles of pH that show a near surface maximum and low subsurface values leading to pH excursions > 1 pH unit are described as indicative of spatially separated redox half reactions mediated by cable bacteria. The authors note in section 4.1.4 a pH profile from Santa Monica basin measured by Reimers et al. (1996) has this form. However, they say little more about this location or the profile. Several studies have characterized Santa Monica Basin porewater chemistry including a paper by Jahnke, R. A. (1990) Early diagenesis and recycling of biogenic debris at the seafloor, Santa Monica Basin, California. J. Mar. Res 48:413-436. Of note, bottom water pH values are relatively low in Santa Monica Basin and so is the bottom water dissolved oxygen concentration. This may be why the pH excursion observed is only about 0.4 pH units. It would seem that sites where the overlying water is in equilibrium with atmospheric oxygen and atmospheric CO2 are more poised to produce the large pH excursions described by the authors.

**Response**

This suggestion by the referee is highly valuable. pH excursions indeed seem to be dependent on the pH of the overlying water. Lower ΔpH are measured in more acidic bottom waters (as is the case in the Santa Monica basin). We also agree with the reviewer that the description of the

Santa Monica Basin was minimal. We have added some additional text to place this observation in the right context.

**Changes in the text**

(5) Sayama (2011) reported cable bacteria activity in the seasonally hypoxic Tokyo Bay (Japan). Moreover, a pH profile from 900m deep within the Santa Monica basin (USA; Reimers et al., 1996), also indicates the presence of e-SOx. Intriguingly, the Santa Monica basin is an anoxic basin with occasional oxygen inflows, and the pH profile was recorded under low oxygen (8.2 µmol $L^{-1}$ $O_2$) and low pH conditions (pH ~7.5 in the bottom water). The pH excursion recorded within the Santa Monica basin ($\Delta$pH ~ 0.4) was lower than most sites reported here, which could be due to the acidification of the bottom water (Jahnke 1990) modulating the pH fingerprint resulting from e-SOx. Hence, sites with bottom waters in equilibrium with atmospheric oxygen and atmospheric $CO_2$ could be poised to produce larger pH excursions than permanent or seasonally hypoxic basins.

Minor Editorial Suggestions:

We agree with the reviewer on all proposed editorial suggestions and have incorporated them in the text.

Page 1, Line 20: The locales listed here in the abstract are not "oceanographic regions or climate zones". It would be more accurate to say "(coastal sites off The Netherlands, Greenland, the USA and Australia)".

We have incorporated this remark. The text has been changed to "is found in coastal environments within different climate zones (off the Netherlands, Greenland, USA, Australia)"

Page 1, Line 29. Change to "induce electron transport" (drop "an").

We have incorporated this remark

Page 2, Line 4. Change to "electro-active marine sediments hosting cable bacteria."

We changed this to "but that some marine sediment are electro-active"

Page 2, Lines 24-25. Change to "cable bacteria may extend well beyond the sediment and impact the water column chemistry".

We have incorporated this remark

Page 2, Lines 33-34. Change to "only a few reports from a restricted number of coastal sites and habitats describe cable bacteria."

We have incorporated this remark

Page 3, Line 13. Change to "cable bacteria in seafloor sediments". If you were considering "the seafloor" you would be considering all types of substrates including bare rock. Same comment applies on page 4, line 10.

We have incorporated this remark

Page 4, Line 18. Reword as "Metabolic activity in the field can be detected either by in situ microsensor profiling: : :.. either on shipboard: : :"

We have incorporated this remark

Page 4, Line 23. Reword as "Cores from deeper areas were retrieved using a gravity core sampler: : :."

We have incorporated this remark

Page 5, Line 5. Reword as "in a fashion similar to previous studies: : :"

We have incorporated this remark

Page 5, Line 9. Reword as "allows assessment of cable bacteria effects on the: : :."

We have incorporated this remark

Page 6, Line 18. Provide after "the R package marelac" some reference or description to explain what this is, who developed it, and where it is available.

We have incorporated this remark

Page 16, Line 13. Reword as: "The subsequent development of e-SOx can be rapid."

We have incorporated this remark

Page 16, Line 21. The evidence here is only suggestive so best to reword as "however, suggest the presence of cable bacteria: : :."
We have incorporated this remark

Figure 3. Give units for EP.

We have incorporated this remark

**Reviewer 3**

The manuscript describes the world wide distribution of cable bacteria in marine sediments. It's a thorough work and interesting to the readers of the journal.

We thank the reviewer for the comments and suggestions on the manuscript.

Major comments:

1) A major concern is the style of the manuscript. It is written in a very lenghty, prosaic style with numerous repetitions between the materials, results, and discussion parts. The manuscript can be easily shortened to half.

2) The materials, results, and discussion are not well separated but appear very mixed. Lots of materials are presented in the results etc.

**Response:**

The reviewer's main comment concerns the structure of our manuscript, and the advice is to change this structure, so that it strictly conforms to the traditional "materials, results, and discussion" format. We are not convinced that such a reformatting would benefit the manuscript.

This is because our manuscript combines two goals: firstly, we review the existing knowledge on the global distribution of cable bacteria, and secondly, we add a large number of new observations of cable bacteria. While the second aspect could be well covered by a traditional "materials, results, and discussion" format, this is not so much the case for the review part of the manuscript.

Our decision on the structure of the manuscript was not taken lightly. We have tested various structures for the paper and finally decided on the current format, as this structure benefits the reading flow the most and thus improves the readability of the paper. Considering that the two other reviewers did not provide any comments or objections, we decided to keep the current structure.

Still, we agree that repetitions were present in the text, as pointed out by the reviewer, and we have modified the text as indicated below. Overall, we have carefully reassessed the manuscript text based on the comments of the reviewer, and the resulting modifications have substantially improved the text.

3) The manuscript leaves the impression that LDET is the most dominant type of sulfur metabolism in marine sediments which might be true or not. It should be compared to the classical redox sequence model which has been found almost everywhere until recently and the overall importance of the two models should be discussed.

**Response**

This is a valuable comment, which pinpoints an important (missing) nuance in our conclusions. Effectively, it was not our intention to suggest that e-SOx is the most dominant type of sulfur oxidation in marine sediments. The aim of this paper is to summarize the present knowledge of on the distribution of e-SOx. We show that e-SOx is likely globally present in coastal areas,

indicating that it could be *an* important sulfur oxidation mechanism (which however does not imply *the most* important). We cannot make a ranking of the importance, because there is no similar data available on the abundance and distribution of the other sulfur oxidation mechanisms (which are less distinctive in their geochemical fingerprint and so more difficult to detect)

The overall importance of e-SOx compared to other sulfur metabolisms is a very interesting question and is (clearly) a research direction for future research. To clarify this point we have rewritten the conclusion of the paper (6).

**Changes in the text**

(6) Conclusion P20 L5 – L8

The combination of a widespread occurrence and a strong local geochemical imprint suggests that cable bacteria could be important in the cycling of carbon, sulfur and other elements in coastal environments. Future research should focus on quantifying the respective share of e-SOx in the natural environment compared to the more classical redox sulfide oxidation. The seasonality and spatial heterogeneity of cable bacteria occurrence are still poorly understood in most habitats.

4) I suggest to not use the term eSOX. The electricity component is added by humans. The bacs don't generate electricity. The fact that humans can measure an electric field under very high activities and high resistance of the sediment does not mean that an electric field is always generated in nature. It's a very catchy term but not scientifically correct in my eyes.

**Response:**

Electrogenic sulfur oxidation (abbreviated e-SOx) is the generally accepted term used by all groups working within the research community on LDET. All papers dealing with cable bacteria are now systematically using this terminology.

The rationale behind the term "electrogenic" is to differentiate between sulfide oxidation over long-distance electron transport and "classical" sulfide oxidation. Electricity is defined as electron transport through a medium. In both metabolisms, electricity is generated by bacteria, even though the scale is very different. In classical sulfide oxidation, electrons are moved over nanometer distances across the enzymes of the electron transport chain. However, in the case of cable bacteria, the electron transport is scaled to centimetre distances.

5) The data can be presented in a more systematic way like in table 1. I would not present all the detailed figures which all show the same content. This would also contribute to a drastic shortening of the manuscript.

**Response:**

We thank the reviewer for this suggestion. All data has been summarized in table S1, which is given as supplemental information to this paper. However, we do feel the figures are illustrative, as they show the variety in geochemical signatures that are observed in the natural environment

as well as the variety in habitats with cable bacteria. Therefore we have retained them within the manuscript.

Specific comments:

6) Page 5, line 17-30: repetition

**Response:**

While the presence of a distinctive geochemical signature during cable bacteria activity is mentioned in the introduction, these sentences explain what features in the geochemical depth profiles of $O_2$, $H_2S$ and pH compose the geochemical signature. Therefore, we have retained this paragraph but shortened it.

**Changes in the text (P5; L17)**

(7) The metabolic activity of cable bacteria imposes a distinct geochemical fingerprint on the porewater, which is revealed by a combination of oxygen ($O_2$), free sulphide ($H_2S$) and pH microsensor profiling (Nielsen et al. 2010; Meysman et al. 2015). The first characteristic feature is the presence of a wide suboxic zone (here: [$H_2S$] and [$O_2$] < 1 µmol $L^{-1}$). The depth of the suboxic zone is a good indication of the depth to which the cable bacteria network is present (Schauer et al. 2014). Secondly, the spatial segregation of the two redox half-reactions leads to a distinct pH signature in the sediment porewater (Fig. 1; Meysman et al., 2015). The anodic oxidation of sulfide at depth causes proton production, whilst conversely, the cathodic reduction of oxygen and/or nitrate in the top sediment leads to strong proton consumption.

7) P6, l 8-11: repetition

**Response:**

We agree with the reviewer that the introduction mentions the built-up of an electrical field in the sediments by cable bacteria. However, the electrical potential profiling method is new in microsensor techniques (see comments reviewer 2). In concert with remark 3 of reviewer 2 we have changed the text as indicated above.

8) P6, l 29-P7, l4: should be in discussion

**Response:**

In this part of the material & methods we explain how we differentiate between cable bacteria and other filamentous bacteria through classical and electron microscopy. We thus still feel these sentences are best placed within the methods at their current position.

9) P 7, l 8: what is modified precisely?

**Response:**

Pasteur pipet tips were elongated using a flame and carefully bent as to form the described fish-hook. To better explain this we have added the following to the text

**Changes in the text**

> (8) Sometimes, cable bacteria were also picked out of the sediment as clumps. For this purpose Pasteur pipettes tips were carefully elongated using a flame and the pipette tip was bent into a U-shape, thus create a "fishing hook".

10) P8, l 12-17: repetition

**Response:**

We suspect something went wrong in the numbering. We assume this comment relates to P7 l 12-17

These sentences introduce the fluorescence *in situ* hybridization technique used in cable bacteria research. Although the technique is mentioned to exist in 2.1, here we provide the full specifications of the procedure that was used.

11) P8, l 24-31: should be in discussion

We suspect something went wrong in the numbering. We assume the comment is about P7 L24-31

**Response:**

See our response above to the request for a major restructuring of the manuscript.

The lines mentioned here introduce the necessity of a proof-of-concept between lab and field measurements and the ensuing results. We have shortened this section to (9).

**Changes in the text (P8, L24)**

> (9) Laboratory incubations of sediments have previously been used as a simple and fast screening technique for potential cable bacteria activity (e.g. Larsen et al. 2015; Burdorf et al. 2016). Previous studies have shown that geochemical fingerprints obtained during laboratory induction are comparable to those obtained from direct field observations (e.g. Seitaj et al. (2015); Rao et al. (2016)). However, in these studies, the sediment was never collected at exactly the same time and place. Here, we performed a direct comparison between laboratory inductions and field measurements. In this methodological test, we compared both the geochemical as well as the electrical fingerprints.

12) P9, l 1-9: should be in Materials and is a repetition

We assume the comment is about P8 L1-9.

**Response:**

See our response above to the request for a major restructuring of the manuscript. Due to the large number of field sites, the reader would be lost without proper context on these sites when

presenting the results. To improve the readability, we therefore introduce the sites in the Results rather than in the Methods.

13) P9, l 11: I suggest to need abbreviate OPD. Makes text more difficult to read.

We assume the comment is about P8 L11

**Response:**

We agree with the reviewer and have replaced the abbreviation OPD with oxygen penetration depth (10).

**Changes in the text**

    (10) Both cores showed a shallow oxygen penetration depth (resp. 0.4 ± 0.06 mm in the field versus 1.2 ± 0.1 mm in the laboratory induction) and a clear acidification of the suboxic zone ($\Delta pH$ = 2.1 ± 0.4 in the field versus $\Delta pH$ = 2.4 ± 0.1 in the laboratory induction).

14) L 12: what delta pH? Delta between what values?

We assume the comment is about P8 L12

**Response:**

$\Delta pH$ is an indicator of the activity of cable bacteria in the sediment and is defined as the maximum pH in the oxic zone minus the minimum pH in the suboxic zone. The definition of $\Delta pH$ is given in Methods section 2.4 about microsensor profiling (P5 L28 – L30).

15) L 14: replace oxygen penetration depth by oxic zone

We assume the comment is about P8 L14

**Response:**

We have replaced the sentence (11).

**Changes in the text**

    (11) The expected pH maximum in the oxic zone was only detectable in the laboratory induction.

16) L 15-22: This comparison cannot be done because the gradients and the zones change over time. Even for lab incubations such a comparison would only be feasible if the profiles are measured at distinct times and if it were shown before that the generation of the profile at a certain time is reproducibly and predictable. However, these gradients are dynamic and even

collapse after a certain time. The situation is even worse in the field, as the kinetics of the profile formation are not known and the time point of sampling within such a built up or collapse time period is not defined.

We assume the comment is about P8 L15 - L22

**Response:**

We agree with the reviewer that geochemical depth profiles of $O_2$, pH and $H_2S$ in sediments are dynamic and thus not always easy to compare. However, in the case of cable bacteria activity, three published papers (Schauer et al. 2014; Vasquez-Cardenas et al. 2015; Rao et al. 2016) have followed the establishment of the geochemical signature by cable bacteria in sediment cores. The features that we compare in this section have been shown to closely follow cable bacteria activity ($\Delta$pH, depth of suboxic zone, electrical potential).

17) L 24-30: Materials
18) P9, l 2-14: Materials

**Response to 17 and 18:**

See our response above to the request for a major restructuring of the manuscript and also remark 12. We introduce the sites here because this makes the manuscript easier to read. However we agree with the reviewer that some repetition is present in this section, so we have revised the following sentences (12):

**Changes in the text (**P9 L2-L5):

    (12) The sites in the Rhine-Meuse-Scheldt delta were located within a geographically restricted area (all within 100 km), but covered a diverse range of coastal habitats (Fig. 4b). We surveyed sediments in three water bodies within the Rhine-Meuse-Scheldt delta, which each have a distinct ecology and hydrodynamics regime: (1) Lake Grevelingen, (2) the Eastern Scheldt and (3) the Western Scheldt.

19) L 28: repetition

We have removed this information from P9 L2-L5

20) L 32-33: delete sentense

We agree with the reviewer and have removed this sentence

21) From here on I stopped indicating that parts belong to Materials or other sections

22) P13, l 9: crucial player: what is crucial? Any estimates of the quantitative importance?
23) L 10: marine sediments: is that only coastal or all?
24) L 8-19: repetition

**Response to 22,23,24:**

We agree with the reviewer that the choice of the word "crucial" was not the best in this context. We have replaced it with "important". We would argue this on the basis of (1) the large share of

cable bacteria on the oxygen uptake of the sediment (2) the built-up of a suboxic zone in the sediment alongside the development of cable bacteria and (3) the mobilization of iron, manganese and calcium from acid-sensitive minerals due to the acidification in the suboxic zone.

While point 2 and point 3 are mentioned within the manuscript, we have added some quantitative arguments for point 1 within the text.

These numbers are from laboratory experiments where sediment with an active cable bacteria population was cut horizontally by passing a thin wire at 1 – 2 cm deep through the sediment. The e-SOx process halted and the oxygen penetration depth immediately deepened (Vasquez-Cardenas et al. 2015). Here, in optimal growing conditions, the oxygen uptake decreased by 70% after the cut.

For sediment cores from the field, similar estimations are not available. However, based on the alkalinity profiles of the sediment or the sulfate production rates in the sediment, estimations on cable bacteria oxygen consumption are possible. From *in situ* cores the share of e-SOx in the total oxygen uptake has been estimated to be between 5-40% (Malkin et al. 2014) and in the lab under optimal conditions between 45-74% (resp. (Rao et al. 2016; Larsen et al. 2015)).

We have added coastal to the text and have changed the text as (13).

**Changes in the text (P13, L9 -)**

(13) Ever since the geochemical signature of cable bacteria was first discovered in 2010 in a laboratory induction experiment (Nielsen et al. 2010), more and more evidence has accumulated that cable bacteria could be important players in the natural elemental cycling of coastal marine sediments (Malkin et al. 2014; Nielsen and Risgaard-Petersen 2015). When cable bacteria are present and active in a given sediment environment, they exert a large impact on the geochemical cycling due to their strong impact on the pH depth profile (Risgaard-Petersen et al. 2012; Rao et al. 2016). Additionally, the share of e-SOx in the total oxygen uptake of the sediment in laboratory incubations can be up to 80% (Vasquez-Cardenas et al. 2015), while field measurements have reported a share in the oxygen uptake of up to 34% (Malkin et al. 2014). Figure 8 provides a global overview of cable bacteria presence and activity, summarizing the new observations presented in this study together with previous reported data from literature (including gene archive data with a >97% similarity to cable bacteria and published geochemical profiles with an e-SOx signature that were not interpreted as such at the time of publication). In the following paragraphs, we provide a short discussion of the different locations and habitats that harbour cable bacteria.

25) L 23: fine-grained: data on sediment properties lacking in Materials part. A characterization of the sediments of all sites would be good.

We agree with the reviewer that it would be good to give a characterization of the sites. Although the information is not available for all sites sampled here, we have now included a table with the porosity, organic C content and grain size for the following sites: Greenland, Long Island Sound, Yarra, Dutch Delta (Texel barrier island (2 sites), Eastern Scheldt (1 site), Grevelingen (1 site) ).

26) P19, l 6-18: This is obviously a new experiment which should be described in materials and in results accordingly.
31) Fig. 9: New experiment –> materials? Results?

The reviewer is correct that the observations presented are not described in the results. However, we believe that these observations fit better into the discussion. In this manuscript we mostly provide a systematic inventory of the occurrence of cable bacteria in coastal marine sediments. The measurements in the bioturbated tank are from an area discussed in 3.2 and the sampling procedure follows the procedure described in section 2.2. These measurement are thus a laboratory control to the possibility of cable bacteria to coexist with bioturbation in sediments.

27) P 19: Conclusion is more a second abstract than a conclusion.

We have amended the conclusion to (14)

**Changes in the text:**

(14) Cable bacteria are globally present and active in marine sediments. Within coastal areas, cable bacteria are active in a variety of habitats (mud flats, seagrass beds, mangroves, salt marshes, seasonally hypoxic basins) and across all latitudes (from tropical to polar environments). Sampling in a restricted geographical area (The Netherlands) revealed that cable bacteria are found in many locations and at different time points in these coastal habitats. This high ratio of successful detection relative to sampling effort suggests a widespread occurrence. We contend that similar sampling campaigns in other coastal locations would reveal a similar common occurrence of cable bacteria activity. Previous studies have documented a considerable impact of e-SOx on the elemental cycling (Risgaard-Petersen et al. 2012) and fluxes across the sediment-water interface (Rao et al. 2016). This combination of a widespread occurrence and a strong local geochemical imprint suggests that cable bacteria could be important in the cycling of carbon, sulfur and other elements in coastal environments. Future research should focus on quantifying the respective share of e-SOx in the natural environment compared to the other types of sulfide oxidation. Seasonality and spatial heterogeneity of cable bacteria are still poorly understood in most habitats. Furthermore, the deep sea remains largely unexplored with respect to the presence and activity of cable bacteria. Based on the scarce evidence that is currently available, cable bacteria seem to be restricted to localized hotspots where sulfide is in sufficient supply (deep sea fans, hot vents, cold seeps).
This study also provides a better insight to the environmental constraints on the natural distribution of cable bacteria within the seafloor sediments. Cable bacteria are capable of growing in a wide range of temperature and salinity conditions. The ecological niche of cable bacteria appears to be primarily constrained by the availability of suitable electron acceptors ($O_2$/$NO_3$) and the availability of free sulfide as the electron donor (liberated in the porewater through sulfate reduction or through the dissolution of FeS). Sediment disturbance through bioturbation appears to mechanically impede the filament network of cable bacteria, but fast regrowth appears to occur when a sediment patch is left undisturbed for a sufficiently long period. This observation that cable bacteria are also present in bioturbated sediments greatly extends their potential distribution in the present-day seafloor sediments (where most sediments are bioturbated).

28) Figure 1: unnessassary abbreviations are introduced which are not used at any other place in the manuscript.

We agree with the reviewer and have removed the abbreviations COR and ASO from the figure.

29) Figure 5: name sites (1-4) the same as the panels (A-D)

We have changed the caption of this figure following the remarks of the reviewer.

30) Fig. 5: Indications of presense seems arbitrary. I suggest presenting the data in a table to make it comparable. Electron micrographs do not show ridges here!

We have added reference to the supplemental information table with all the data. Electron microscopy images have been changed or further zoomed into to show the ridges of the cable bacteria (see also comment by reviewer 2). Full-size figures are added as supplemental information to the paper.

**References**

Burdorf, L. D. W., S. Hidalgo-Martinez, P. L. M. Cook, and F. J. R. Meysman. 2016. Long-distance electron transport by cable bacteria in mangrove sediments. Mar. Ecol. Prog. Ser. **545**: 1–8. doi:10.3354/meps11635

Damgaard, L. R., N. Risgaard-Petersen, and L. P. Nielsen. 2014. Electric potential microelectrode for studies of electrobiogeophysics. J. Geophys. Res. Biogeosciences **119**: 1906–1917. doi:10.1002/2014JG002665

Jahnke, R. A. 1990. Early diagenesis and recycling of biogenic debris at the seafloor, Santa Monica Basin, California. J. Mar. Res. **48**: 413–436.

Larsen, S., L. P. Nielsen, and A. Schramm. 2015. Cable bacteria associated with long-distance electron transport in New England salt marsh sediment. Environ. Microbiol. Rep. **7**: 175–179. doi:10.1111/1758-2229.12216

Malkin, S. Y., A. M. Rao, D. Seitaj, D. Vasquez-Cardenas, E.-M. Zetsche, S. Hidalgo-Martinez, H. T. Boschker, and F. J. Meysman. 2014. Natural occurrence of microbial sulphur oxidation by long-range electron transport in the seafloor. ISME J. **8**: 1843–1854. doi:10.1038/ismej.2014.41

Meysman, F. J. R., N. Risgaard-Petersen, S. Y. Malkin, and L. P. Nielsen. 2015. The geochemical fingerprint of microbial long-distance electron transport in the seafloor. Geochim. Cosmochim. Acta **152**: 122–142. doi:10.1016/j.gca.2014.12.014

Nielsen, L. P., and N. Risgaard-Petersen. 2015. Rethinking sediment biogeochemistry after the discovery of electric currents. Annu. Rev. Mar. Sci. **7**: 425–442. doi:10.1146/annurev-marine-010814-015708

Nielsen, L. P., N. Risgaard-Petersen, H. Fossing, P. B. Christensen, and M. Sayama. 2010. Electric currents couple spatially separated biogeochemical processes in marine sediment. Nature **463**: 1071–1074. doi:10.1038/nature08790

Rao, A. M. F., S. Y. Malkin, S. Hidalgo-Martinez, and F. J. R. Meysman. 2016. The impact of electrogenic sulfide oxidation on elemental cycling and solute fluxes in coastal sediment. Geochim. Cosmochim. Acta **172**: 265–286. doi:10.1016/j.gca.2015.09.014

Reimers, C. E., K. C. Ruttenberg, D. E. Canfield, M. B. Christiansen, and J. B. Martin. 1996. Porewater pH and authigenic phases formed in the uppermost sediments of the Santa Barbara Basin. Geochim. Cosmochim. Acta **60**: 4037–4057.

Revil, A., C. A. Mendonça, E. A. Atekwana, B. Kulessa, S. S. Hubbard, and K. J. Bohlen. 2010. Understanding biogeobatteries: Where geophysics meets microbiology. J. Geophys. Res. Biogeosciences **115**: G00G02. doi:10.1029/2009JG001065

Risgaard-Petersen, N., L. R. Damgaard, A. Revil, and L. P. Nielsen. 2014. Mapping electron sources and sinks in a marine biogeobattery. J. Geophys. Res. Biogeosciences **119**: 1475–1486. doi:10.1002/2014JG002673

Risgaard-Petersen, N., M. Kristiansen, R. B. Frederiksen, and others. 2015. Cable Bacteria in Freshwater Sediments. Appl. Environ. Microbiol. **81**: 6003–6011. doi:10.1128/AEM.01064-15

Risgaard-Petersen, N., A. Revil, P. Meister, and L. P. Nielsen. 2012. Sulfur, iron-, and calcium cycling associated with natural electric currents running through marine sediment. Geochim. Cosmochim. Acta **92**: 1–13. doi:10.1016/j.gca.2012.05.036

Sayama, M. 2011. Seasonal dynamics of sulfide oxidation processes in Tokyo Bay dead zone sediment. **Goldschmidt Conference Abstracts**.

Schauer, R., N. Risgaard-Petersen, K. U. Kjeldsen, J. J. Tataru Bjerg, B. B. Jørgensen, A. Schramm, and L. P. Nielsen. 2014. Succession of cable bacteria and electric currents in marine sediment. ISME J. **8**: 1314–1322. doi:10.1038/ismej.2013.239

Seitaj, D., R. Schauer, F. Sulu-Gambari, S. Hidalgo-Martinez, S. Y. Malkin, L. D. W. Burdorf, C. P. Slomp, and F. J. R. Meysman. 2015. Cable bacteria generate a firewall against euxinia in seasonally hypoxic basins. Proc. Natl. Acad. Sci. U. S. A. **112**: 13278–13283. doi:10.1073/pnas.1510152112

Trojan, D., L. Schreiber, J. T. Bjerg, A. Bøggild, T. Yang, K. U. Kjeldsen, and A. Schramm. 2016. A taxonomic framework for cable bacteria and proposal of the candidate genera Electrothrix and Electronema. Syst. Appl. Microbiol. **39**: 297–306. doi:10.1016/j.syapm.2016.05.006

Vasquez-Cardenas, D., J. van de Vossenberg, L. Polerecky, and others. 2015. Microbial carbon metabolism associated with electrogenic sulphur oxidation in coastal sediments. ISME J. **9**: 1966–1978. doi:10.1038/ismej.2015.10